# Structural basis for the prion-like MAVS filaments in antiviral innate immunity

Hui Xu[1,2], Xiaojing He[3], Hui Zheng[1], Lily J Huang[1], Fajian Hou[2†], Zhiheng Yu[4], Michael Jason de la Cruz[4], Brian Borkowski[1], Xuewu Zhang[3,5*], Zhijian J Chen[6*], Qiu-Xing Jiang[1*]

[1]Department of Cell Biology, University of Texas Southwestern Medical Center, Dallas, United States; [2]Department of Molecular Biology, University of Texas Southwestern Medical Center, Dallas, United States; [3]Department of Pharmacology, University of Texas Southwestern Medical Center, Dallas, United States; [4]CryoEM Shared Resources, Howard Hughes Medical Institute, Janelia Farm Research Campus, Ashburn, United States; [5]Department of Biophysics, University of Texas Southwestern Medical Center, Dallas, United States; [6]Department of Molecular Biology, Howard Hughes Medical Institute, University of Texas Southwestern Medical School, Dallas, United States

*For correspondence: xuewu. zhang@utsouthwestern.edu (XZ); zhijian.chen@utsouthwestern.edu (ZJC); qiu-xing.jiang@ utsouthwestern.edu (Q-XJ)

†Present address: State Key Laboratory of Cell Biology, Institute of Biochemistry and Cell Biology, Shanghai Institutes for Biological Sciences, Chinese Academy of Sciences, Shanghai, China

**Abstract** Mitochondrial antiviral signaling (MAVS) protein is required for innate immune responses against RNA viruses. In virus-infected cells MAVS forms prion-like aggregates to activate antiviral signaling cascades, but the underlying structural mechanism is unknown. Here we report cryo-electron microscopic structures of the helical filaments formed by both the N-terminal caspase activation and recruitment domain (CARD) of MAVS and a truncated MAVS lacking part of the proline-rich region and the C-terminal transmembrane domain. Both structures are left-handed three-stranded helical filaments, revealing specific interfaces between individual CARD subunits that are dictated by electrostatic interactions between neighboring strands and hydrophobic interactions within each strand. Point mutations at multiple locations of these two interfaces impaired filament formation and antiviral signaling. Super-resolution imaging of virus-infected cells revealed rod-shaped MAVS clusters on mitochondria. These results elucidate the structural mechanism of MAVS polymerization, and explain how an α-helical domain uses distinct chemical interactions to form self-perpetuating filaments.

## Introduction

Viral infection of host cells triggers innate and adaptive immune responses that are essential for the survival of the host (*Iwasaki and Medzhitov, 2010*; *Ronald and Beutler, 2010*; *Takeuchi and Akira, 2010*). Initiation of innate immune response relies on a group of pattern recognition receptors, which recognize specific pathogen-associated molecular patterns (PAMPs), including microbial nucleic acids, bacterial cell wall components and certain highly conserved proteins. Two groups of pattern recognition receptors are responsible for detecting viral RNAs. The first group is the membrane-anchored Toll-like receptors (TLRs), located on the cell surface or in endosomal membranes and acting as sensors of viral RNAs encountered in extracellular environment (*Kawai and Akira, 2006*). The second group of sensors resides in the cytosol. They belong to a family of cytosolic RNA helicases called RIG-I-like receptors (RLRs), including retinoic acid inducible gene-I (RIG-I), melanoma differentiation-association gene 5 (MDA5) and laboratory of genetics and physiology 2 (LGP2) (*Yoneyama and Fujita, 2009*). Recognition of viral RNAs by RLRs leads to the activation of mitochondrial antiviral signaling protein (MAVS; also known as IPS1, VISA and CARDIF). Activated MAVS triggers rapid production of type I

**eLife digest** When infected by a virus, the body will generally launch an immune response to eliminate the infectious agent. Activation of the innate immune system–the first line of defense against infection—requires the host cells to recognize the presence of a pathogen and to sound the alarm once the invader is detected.

Viruses can contain DNA or RNA, and when a virus containing double stranded RNA enters a cell, or starts replicating within the cytoplasm, proteins called RIG-I-like receptors (RLRs) will detect these RNA molecules. This will trigger a signaling cascade that results in the production of type I interferons, the proteins that activate cells of the innate immune system.

Members of the RLR family of receptors, including RIG-I and MDA5, initiate the signaling cascade by interacting with the mitochondrial antiviral-signaling (MAVS) protein. Recent work revealed that upon activation by RIG-I or MDA5, MAVS proteins aggregate on the surface of mitochondria and form protein filaments. These filaments then activate inactive MAVS proteins, leading to the formation of more filaments. While a region of the MAVS protein called caspase activation and recruitment domain (CARD) is known to be involved in the formation of the filaments, the chemical interactions that govern the formation process have yet to be described.

Now, using cryo-electron microscopy, Xu et al. have shown that these filaments are comprised of three-stranded helixes. This came as something of a surprise because other similar filaments known as prions are made of tightly packed beta sheets. Xu et al. went on to visualize full-length MAVS filaments in virus-infected cells, and to verify that mutations that impair the assembly of MAVS filaments also prevent RNA viruses from triggering the production of interferon. These results have the potential to inform future studies of the innate immune response, as well as investigations into the assembly of proteins to form prion-like filaments.

interferons and proinflammatory cytokines (*Kawai et al., 2005*; *Meylan et al., 2005*; *Seth et al., 2005*; *Xu et al., 2005*).

RIG-I and MDA5 are two DExD/H-box helicases that belong to the superfamily 2 of RNA helicases (*Yoneyama et al., 2004*; *Fairman-Williams et al., 2010*). Both proteins contain N-terminal tandem caspase activation and recruitment domains (CARDs), a central RNA helicase, and a C-terminal regulatory domain (CTD). Despite sharing a similar domain structure, RIG-I and MDA5 are activated by complementary sets of viral RNA ligands through distinct mechanisms (*Kato et al., 2008*; *Loo et al., 2008*; *Iwasaki, 2012*). RIG-I recognizes short blunt ends of dsRNA with 5′-triphosphate caps (*Hornung et al., 2006*; *Schlee et al., 2009*), as well as long dsRNAs (*Kohlway et al., 2013*; *Patel et al., 2013*; *Peisley et al., 2013*). Ligand binding to the helicase domain and CTD induces a conformational change that liberates RIG-I from an autoinhibited state and exposes its N-terminal tandem CARDs (*Hou et al., 2011*; *Kowalinski et al., 2011*; *Luo et al., 2011*). In contrast, MDA5 detects long dsRNAs made of hundreds to thousands of base pairs (*Kato et al., 2008*). The helicase domain and CTD of MDA5 cooperatively assemble into helical filaments along the dsRNA, leaving the tandem CARDs of MDA5 flexibly exposed on the periphery of the filament (*Peisley et al., 2011*; *Berke and Modis, 2012*; *Wu et al., 2013*). The exposed CARDs of RIG-I and MDA5 subsequently bind to unanchored lysine-63 (K63) polyubiquitin chains and form oligomers. The latter gain a high capacity of activating MAVS on mitochondria, presumably through CARD–CARD interactions (*Zeng et al., 2010*; *Jiang et al., 2012*). The exact mechanism for the polyubiquitin-dependent interaction between active RIG-I and inactive MAVS is not well defined.

MAVS is ubiquitously expressed on the outer membrane of mitochondria. It consists of an N-terminal CARD domain, a proline-rich region (PRR) preceding a poorly structured middle segment, and a monotopic transmembrane (TM) domain at the very C-terminus (*Figure 1A*). Recent studies have shown that upon activation, MAVS molecules polymerize themselves into functional aggregates (*Hou et al., 2011*). These high molecular weight aggregates behave like prion fibers, because they are detergent-resistant, protease-resistant and self-perpetuating by inducing inactive MAVS to form functional aggregates. The N-terminal CARD domain of MAVS is necessary and sufficient for forming active MAVS aggregates. It shares some homology with the first CARD domains in both MDA5 and RIG-I (25% and 20% sequence identity respectively). A crystal structure of the MAVS CARD,

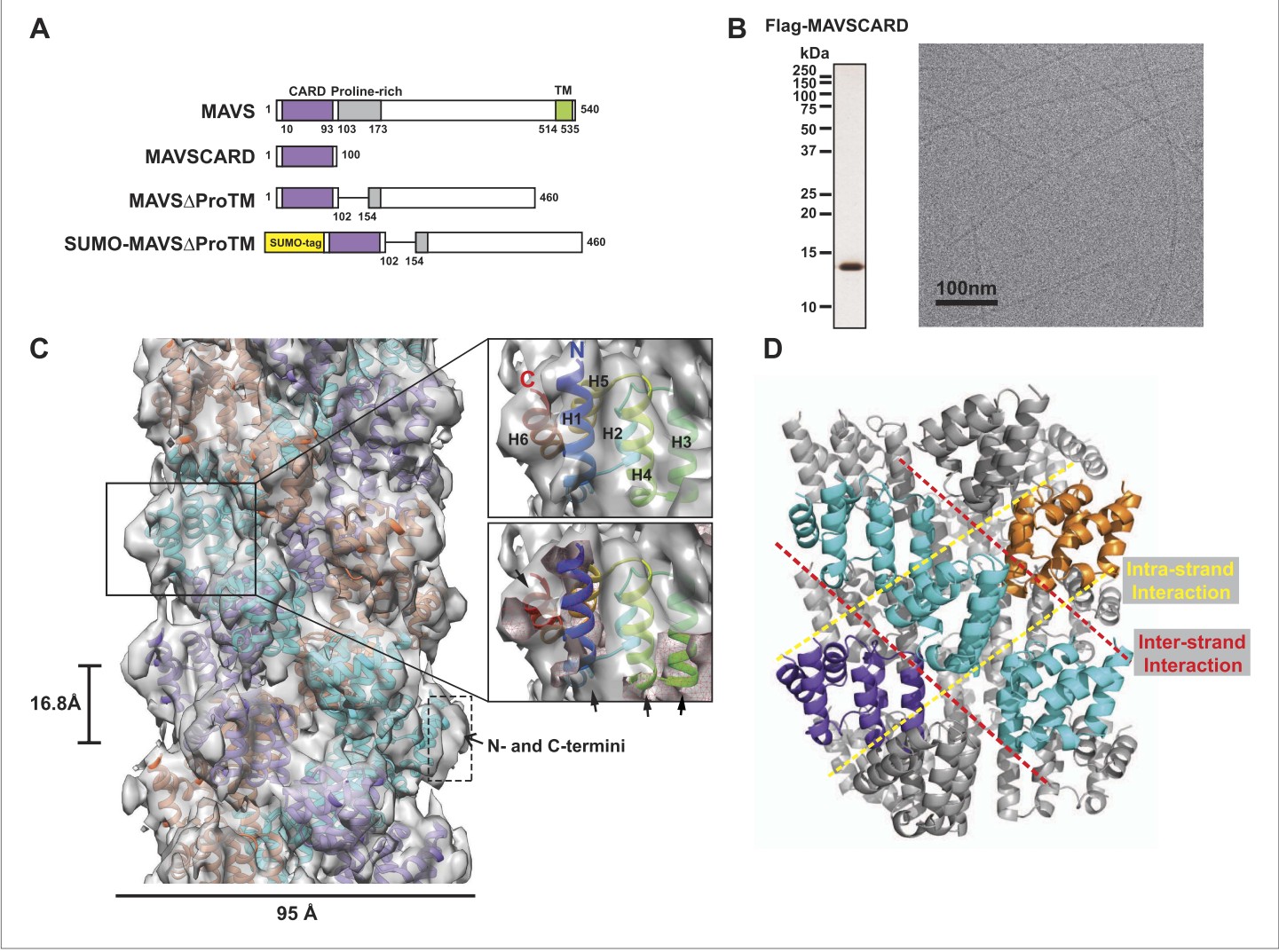

**Figure 1**. CryoEM reconstruction of MAVS CARD filaments. (**A**) Diagrams of the domain organization in MAVS and deletion mutants used in this study. (**B**) Flag-MAVS CARD purified from HEK293T cells analyzed by silver stained SDS-PAGE and cryoEM imaging. The cryoEM image is displayed in reversed contrast (protein in black) for better visualization. (**C**) Left: a side view of the final 3D reconstruction of MAVS CARD filament. The X-ray crystal structure model of the human MAVS CARD (PDB: 2VGQ) was docked into the cryoEM map. Three strands are colored differently. Right: the rod-like densities at the periphery of the cryoEM density map allowed positioning of H1, H4, H3 and H6 without modification (top; see *Video 2*). When a front part of the density map was sectioned off with a clipping plane (red mesh), the H1 helix (blue) fitted into a rod-like density very well (bottom). (**D**) Pseudoatomic model of MAVS CARD filament. Dashed lines indicate the inter- (red) and intra-strand (yellow) interaction interfaces.

The following figure supplements are available for figure 1:

**Figure supplement 1**. Processing of cryoEM images of the MAVS CARD filaments.

**Figure supplement 2**. Structural model of MAVS CARD filaments and its chirality determination by cryo electron tomography (cryo-ET).

fused to a maltose-binding protein (MBP), exhibits a typical helical bundle of six antiparallel α-helices (***Potter et al., 2008***). However, the isolated MAVS CARD without the MBP self-assembles into filamentous structures, which can promote endogenous, inactive MAVS to form highly active aggregates (***Hou et al., 2011***). How the CARD domain triggers MAVS aggregation remains poorly understood.

The unique filamentous structure of MAVS CARD probably results from its distinct chemical properties not shared by other CARDs. To understand the structural basis underlying the filament formation, we

solved the 3D structure of MAVS CARD filaments at 9.6 Å resolution by cryo-electron microscopy (cryoEM) and iterative helical real space refinement (IHRSR) (*Frank, 2006*; *Egelman, 2007*). Based on the cryoEM map and the crystal structure of the individual MAVS CARD, we built a pseudoatomic model of the filament and identified two new CARD–CARD interfaces that are important for filament formation. Mutations of residues found at the two interfaces disrupted MAVS self-association and abrogated the activation of the signaling pathway in cells. In order to understand the domain arrangement of the native MAVS aggregates, we obtained a 16.4 Å cryoEM map of a nearly full-length MAVS protein without part of the proline-rich region (PRR) and the C-terminal transmembrane domain (MAVSΔProTM). The cryoEM map of the MAVSΔProTM filament has the same CARD filament in the center, which is surrounded by extra fragmented densities in the periphery. This arrangement makes the CARD filament the organization center of the MAVS aggregates, and suggests a novel mechanism to expose the central segments of MAVS for downstream signaling effector recognition and signal amplification. To visualize the full-length MAVS filaments in virus-infected cells, we obtained three-dimensional Structured Illumination Microscopic (3D-SIM) images that achieved a sufficiently high resolution for us to discern the rod-shaped ultrathin MAVS filaments. These native thin filaments are on average ~400 nm long and usually seen among mitochondrial membranes. In accordance with the cryoEM studies, point mutations that disrupted MAVS filament formation abrogated the redistribution and aggregation of MAVS on mitochondrial membrane and blocked the induction of interferon-β (IFNβ) in response to RNA virus infection. These results elucidate the structural mechanism for the formation of functional MAVS filaments.

## Results

### MAVS CARD forms three-stranded helical filaments

Our previous electron microscopic (EM) images of negatively stained specimens suggested that the MAVS CARD assembles into a filament-like structure in vitro (*Hou et al., 2011*). To further uncover the molecular mechanism governing the MAVS CARD self-association, we utilized cryoEM to determine the molecular structure of the CARD filament. Flag-tagged MAVS CARD (residues 1–100) was expressed in HEK293T cells and purified to apparent homogeneity (*Figure 1B*). The purified protein formed filaments that eluted from gel filtration column in the void volume. CryoEM images of the purified filaments showed helical diffraction (*Figure 1—figure supplement 1A*, left). We selected good EM micrographs and high-quality Falcon Direct Detector images, and built a large dataset for IHRSR analysis. IHRSR describes the helical symmetry with a general definition of azimuthal rotation (ΔΦ) and axial displacement (Δz) per subunit relative to the helical axis.

Individual datasets from different sessions of data collection were first analyzed separately to confirm that the total power of centered particles in each dataset showed the typical layer lines (*Figure 1—figure supplement 1A*, left). Good datasets showed a clear meridional line at ~16.7 Å (layer line #9), which is a good estimate of Δz. Different datasets were scaled and merged into a large set by their symmetry parameters (Δz; details in the 'Materials and methods'). The final dataset contained 48,884 particles. Multivariate statistical analysis of the images revealed obvious helical properties and imperfection in some of the Eigen images (*Figure 1—figure supplement 1B*). The IHRSR analysis of the dataset started with a featureless cylinder as the initial reference and converged to a stable solution (*Figure 1—figure supplement 1C,D*). After sorting the filaments to consider variations in helical symmetry and accounting for the filaments that may be tilted out of the horizontal plane by up to 15°, we calculated a final map from 15,366 boxed segments of filaments. The resolution of the map was estimated to be 9.6 Å from Fourier Shell Correlation (FSC) between two independently calculated 3D reconstructions (FSC = 0.5; *Figure 1C*, *Figure 1—figure supplement 1E,F*; and *Video 1*). The cryoEM map shows a three-stranded helical assembly with a central pore that is about 18 Å in diameter. Neighboring subunits in each strand are related by an azimuthal rotation angle (ΔΦ) of 53.6° and an axial rise (Δz) of 16.8 Å along the helical axis (*Figure 1C*).

In order to determine the handedness of the helical assembly of MAVS CARD, we obtained cryo-electron tomograms (cryo-ET) of the CARD filaments ('Materials and methods'). *Figure1—figure supplement 2B* shows a 2.7 nm-thick slice of the tomogram as viewed from the outer surface of the filaments. Apparent helical stripes observed in multiple filaments of different orientations suggest that the actual helical structure is left-handed. The measured distance between the helical strands from the

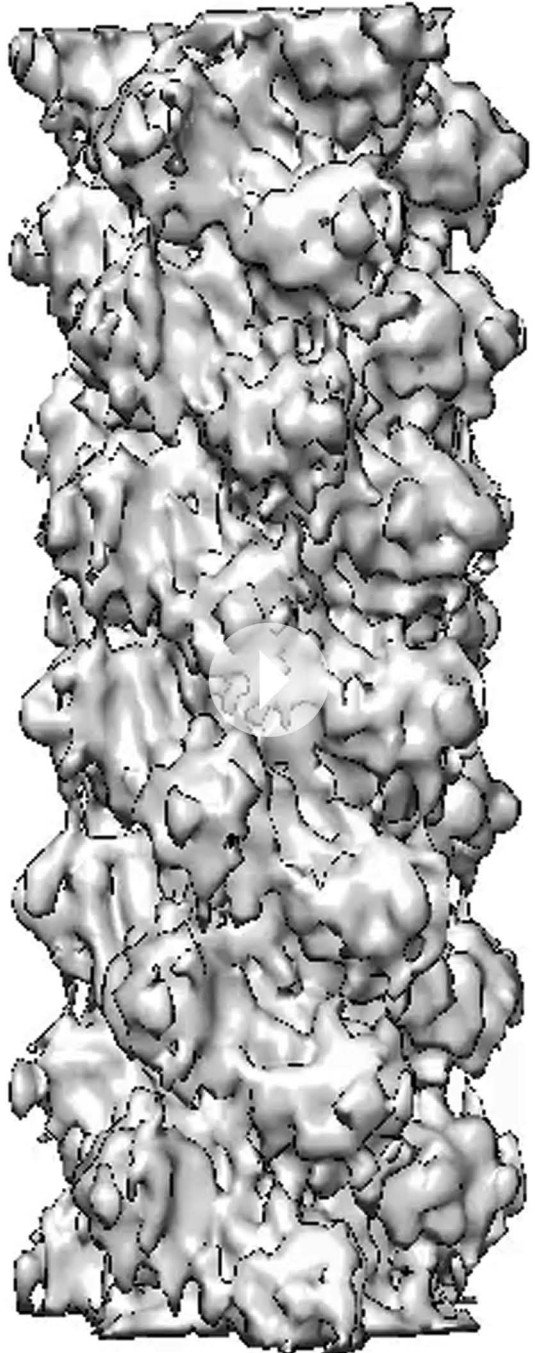

**Video 1**. 3D reconstruction of the MAVS CARD filament.

cryo-ET reconstruction matches well with the cryo-EM reconstruction of MAVS CARD filament (*Figure 1—figure supplement 2A*).

## Pseudo-atomic model of MAVS CARD filament

Although the MAVS filaments share some features with prion fibers, such as self-perpetuation, they could not be stained with Congo red, a dye commonly used to label β sheet-rich insoluble amyloid aggregates formed by most prions (*Hou et al., 2011*). MAVS CARD lacks the glutamine/asparagine-rich regions that are responsible for forming the parallel β-sheet structures of most amyloid fibers (*Michelitsch and Weissman, 2000*; *Nelson et al., 2005*). There is no evidence that during filament formation MAVS CARD undergoes a helix-to-beta-sheet transition in its secondary structure. We therefore built a model of MAVS oligomer by directly fitting the crystal structure of individual MAVS CARD into the cryoEM density map (*Video 2*; *Potter et al., 2008*).

At the peripheral surface of the cryoEM map, three rod-shaped EM densities are clearly discernable (*Figure 1C*; *Video 2*), which are likely contributed by three of the six α-helixes of each CARD molecule. These features enabled us to determine a unique orientation of the crystal structure of the MAVS CARD (residues 1–93) in the cryoEM map. Recognition of these structural features also allowed for independent confirmation of the chirality to the cryoEM map because only the left-handed map allowed the three helices (H1, H4 and H3) to be positioned well into the density map without modification (*Video 2*). After the docking of these three helices, the 6th helix (H6) was naturally fit into a rod-shaped feature next to the H1. Rigid body refinement using SITUS (*Wriggers, 2010*) then locally optimized the agreement between the X-ray model and the cryoEM map (insets in *Figure 1C*; *Video 2*). Even though both H2 and H5 are contained in the density map, there is a small discrepancy between their orientations and the density features lining the inner pore of the filament (*Video 2*), suggesting a possible local rearrangement of the CARD domain upon filament formation (*Figure 1C*).

Because the rearrangement of H2 and H5 is not well defined by the density features and there was a small density next to H5 that is not fully accounted for due to probably inexact segmentation of the helical density map into individual units, we did not use flexible fitting to optimize the local positions of these two helices. Helical symmetry operations and threefold symmetry were subsequently applied to generate a pseudo-atomic model for the CARD filament (*Figure 1C*; *Video 2*).

Overall the MAVS CARD filament adopts a densely packed structure containing three intertwined helical strands. Each turn of one helical strand contains 6.7 (360/53.6) CARD monomers. The first (H1) and last (H6) helices of the CARD are positioned at the outer surface of the filament (*Figure 1C*; *Video 2*). This arrangement provides the structural basis for other domains to be connected to the filaments

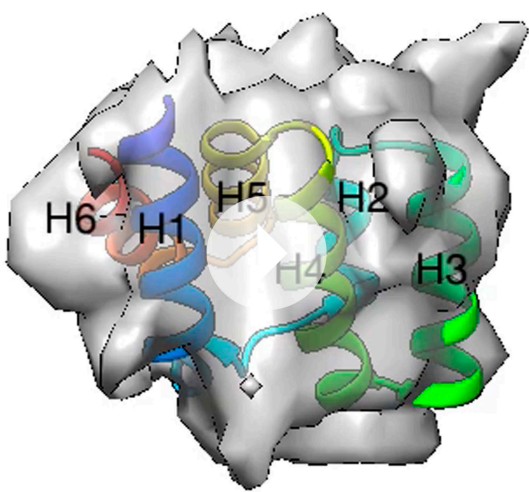

**Video 2**. Building a pseudoatomic model from the cryoEM map. The cryoEM map of the MAVS CARD filament is shown at two different density levels together with the pseudoatomic model, and individual units are assembled into a long filament.

in the center, in keeping with previous observations that a N-terminal small ubiquitin-like modifier (SUMO) tag and other domains added to the C-terminus of CARD did not prevent filament formation (*Hou et al., 2011*). There is some unoccupied density at the surface of the EM map next to the N- and C-termini of the docked CARD model. This may be attributed to the residual density from the N-terminal Flag tag and the additional seven residues at the C-terminus in the protein used for preparing the cryoEM specimens, but not in the crystal structure (*Figure 1C*).

## Chemical basis for the assembly of MAVS CARD filaments

In the structural model of the filament, each MAVS CARD monomer directly interacts with four nearby monomers: two from the same strand and the other two respectively from two adjacent strands (*Figure 1D*). Within one layer perpendicular to the helical axis, the three subunits do not make direct contact with each other. Two types of interfaces are involved in the packing interactions. The intra-strand interface makes contacts between adjacent CARDs within the same strand (*Figure 1D*, *Figure 1—figure supplement 2A*) while the inter-strand interface holds the three strands together.

### The inter-strand interaction interface

At the inter-strand interface, the positive and negative charges are alternately distributed at two opposite ends of each CARD subunit, indicating that the inter-strand interaction is mainly electrostatic (*Figure 2A,B*). The positively charged residues R37 in the 3rd helix (H3), R64 and R65 in the loop between the 4th and 5th helices (H4 and H5) from the CARD in a lower layer (purple in *Figure 2B*) are in close proximity to the negatively charged residues D23 in the loop between the 1st and 2nd helices (H1b and H2) and E26 in the 2nd helix (H2) of another CARD molecule (cyan in *Figure 2B*) in the upper layer. E26, R64 and R65 are highly conserved among MAVS molecules from different species, but D23 and R37 are less so (*Figure 2A*). Replacement of D23 with a histidine residue and R37 with either an asparagine or a serine residue in the MAVS orthologs introduces good H-bonders to these less conserved positions, and may compensate for the lost electrostatic pairs.

To test the energetic contributions of these charged residues to the stability of the filament and the in vivo MAVS signaling activity, we mutated them individually to an alanine residue or residues with reversed charges. The cDNAs encoding these MAVS mutants were transiently expressed in HEK293T cells together with an IFNβ luciferase reporter plasmid. IFNβ induction was measured by a luciferase assay. Compared with the wild-type MAVS, which potently induced IFNβ in a dose-dependent manner, point mutations at multiple locations of the inter-strand interface abolished MAVS activity (*Figure 2C*). The gel-filtration profiles (E26R as an example in *Figure 2—figure supplement 1A*) and the EM images of the negatively stained proteins (*Figure 2—figure supplement 1B*) verified that these mutations impaired the filament formation of the MAVS CARD. X-ray crystallographic studies of several horse MAVS CARD mutants, which were monomeric in solution and yielded well diffracting crystals, showed that they adopt almost exactly the same structure as the wild-type protein (e.g., *Figure 2—figure supplement 1C* for the horse E26R and R64C structures; *Table 1*), suggesting that these mutations did not disrupt the filament by changing the structural fold of each subunit, but instead by altering the interface chemistry.

To test the specificity of the electrostatic interactions at the MAVS CARD assembly interface, we mutated several charged residues outside the interface and assessed their effects on MAVS activity. E70 and R77 in the H5, E80 in the H5-H6 loop, as well as D86 and E87 in the H6 are outside the interfaces (*Figure 2D*). Their replacement with alanine or reversely charged residues still allowed potent induction of IFNβ production in HEK293T cells (*Figure 2C*). In contrast, D40, R41 and R43 are located

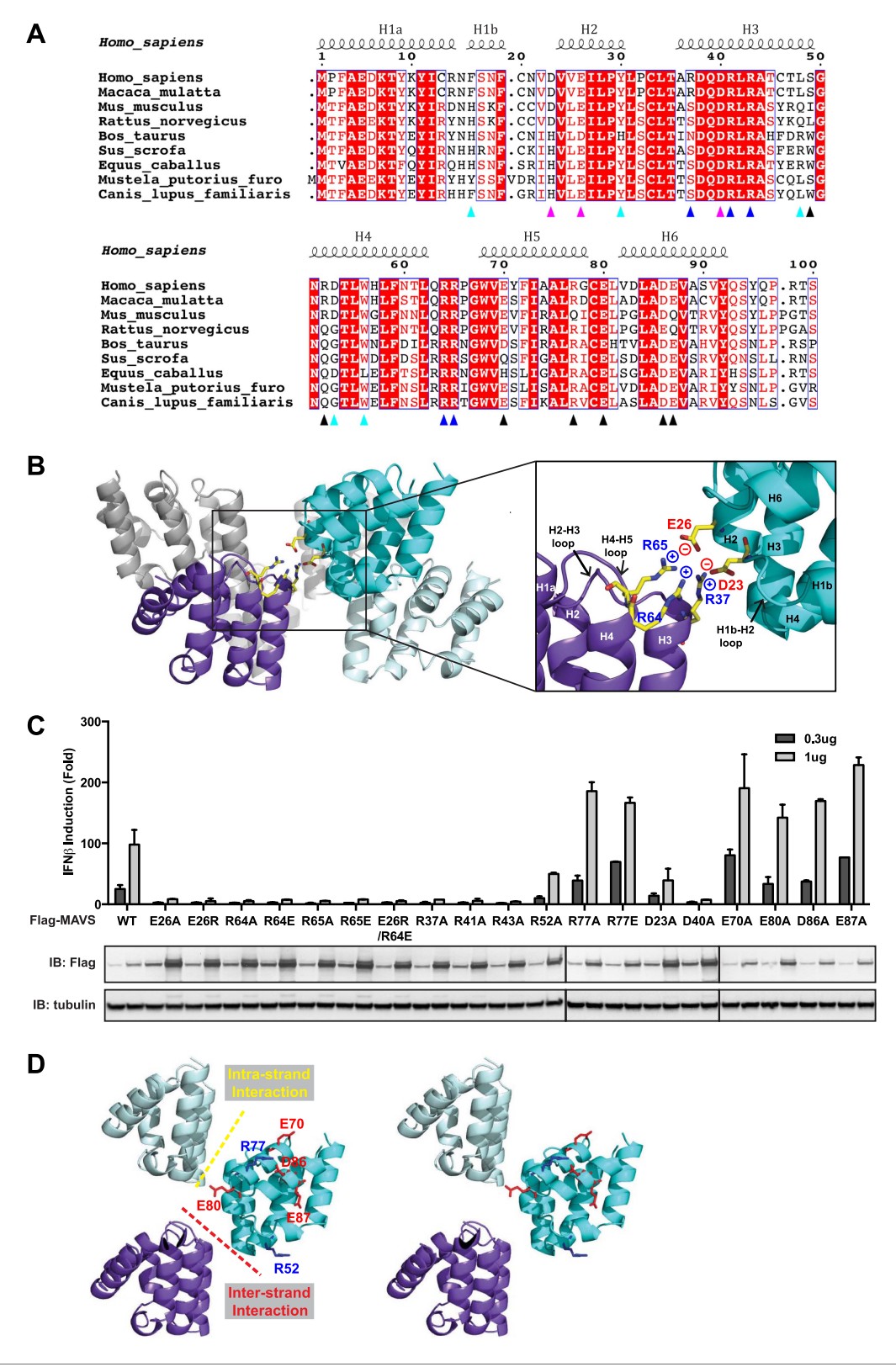

**Figure 2**. Charged residues are conserved at the inter-strand interface of MAVS CARD filament. (**A**) Sequence alignment of the MAVS CARD from 10 different species with the secondary structures based on the X-ray model of human MAVS CARD. The alignment was generated with ClustalW (http://www.ebi.ac.uk/clustalw/) and formatted *Figure 2. Continued on next page*

*Figure 2. Continued*

using ESPript (http://esprit.ibcp.fr/ESPript/ESPript/). The colored arrowheads mark the residues mutated in this study: pink, key negatively charged residues at the inter-strand interface; blue, key positively charged residues at the inter-strand interface; cyan, key residues at the intra-strand interface; black, residues showing normal activity when substituted with alanine. (**B**) MAVS CARD hexamer model with key residues at the inter-strand interface between two subunits (purple and cyan) shown as yellow sticks. The inset on the right side is an expanded view of these residues. The side chains of these residues (D23, E26, R37, R64 and R65) in the model were optimized by testing different rotamers in Coot. (**C**) Effects of mutating the residues at the inter-strand interface and other conserved charged residues on MAVS activity. Wild-type MAVS and CARD mutants at different positions were transiently expressed in HEK293T-IFNβ-luciferase reporter cells. Cells were lysed 24 hr later, and the MAVS signaling was tested by luciferase reporter assay in a dose-dependent manner. Western blot was done to monitor the expression level of the transfected MAVS proteins with α-tubulin as loading controls. (**D**) A stereo view of MAVS CARD model with the surface-exposed, conserved charged residues shown as sticks (blue, positively charged residues; red, negatively charged residues). Mutations of these surface residues, which are not involved in the inter-strand interactions, do not impair MAVS signaling.

The following figure supplements are available for figure 2:

**Figure supplement 1**. Mutations at the inter-strand interface disrupt MAVS CARD polymerization.

in the H3 and very close to the inter-strand interface. Their mutations did exert strong negative effects on MAVS signaling (*Figure 2C*). The exact packing of these three residues and their contribution to the inter-strand interface require better resolution of the cryoEM map.

## The intra-strand interaction interface

The cryoEM model predicts that the intra-strand interface is stabilized mainly by hydrophobic interaction or hydrogen bonds among residues F16, W56, Y30, D53 and L48 (*Figure 3A*). W56 is well conserved (leucine residue in horse). In the crystal structure of MAVS CARD, the side chain of W56 adopts different conformations, with a major conformer (60% occupancy) exposed to solvent and stacked against the ring structure of residue F16 in immediate vicinity (magnified view on the right side of *Figure 3A*) (*Potter et al., 2008*). Y30 is a fairly conserved residue (histidine in cattle) that protrudes to the close proximity of W56 in the other CARD molecule at the interface. Even though the atomic details of the side chain packing are not fully resolved, the three ring structures from these three residues appear to be important for the intra-strand stability. Single alanine substitution of F16, W56 or Y30 almost completely abolished MAVS activity (*Figure 3B*), but mutations of some of the residues next to them (R52 and S49 as examples) had little effect (*Figure 2C*, *Figure 3B*). Interestingly, when we introduced different ring-containing residues to these positions, such as F16H, W56F, W56Y, Y30F and Y30H, we were able to largely rescue MAVS activity (*Figure 3B*). These results support that the hydrophobic packing among the three ring-containing residues is important for the intra-strand interface. D53 is one helical turn below W56, and its side chain sits at the boundary between the aqueous phase and the hydrophobic core packing of F16, W56 and Y30. When we introduced an alanine to this position, it completely abolished the MAVS activity (*Figure 3B*), suggesting that the charged D53 may enforce the hydrophobic interaction above the aqueous surface. Close to the central pore of the helical filament, residue L48 appears in close contact with a group of short chain residues and contributes to a separate hydrophobic patch. Even though atomic details in side chain packing are unclear, substitution of L48 with charged residues (D or K), but not alanine, completely abolished MAVS activity (*Figure 3B*), suggesting that this second hydrophobic patch is also important for intra-strand interaction.

To further test whether these mutations impaired the MAVS signaling activity because they interfered with CARD polymerization, CARD mutants were purified from *Escherichia coli* and their oligomerization states were examined. Unlike the wild-type MAVS CARD, which eluted at the void volume in gel filtration chromatography, W56A, W56E and W56R mutants eluted as monomers (*Table 1*; *Figure 3A*). These results indicate that mutations that abrogate hydrophobic interactions at the intra-strand interface prevent MAVS CARD oligomerization and abolish MAVS activity.

According to our pseudoatomic model, residues in the first helix (H1) are not in direct contact with any adjacent molecule (*Figure 1C*; *Video 2*). This agrees with our previous finding that deletion of the first ten residues in MAVS CARD did not impair its filament formation (*Hou et al., 2011*). In addition, no cysteine pairs are at the interface, which explains the prior observation that MAVS filaments were not disrupted by a high concentration of reducing agent DTT (*Hou et al., 2011*).

**Table 1.** Summary of MAVS CARD mutants

| Residues | Human | | | | Horse | | | |
|---|---|---|---|---|---|---|---|---|
| | Mutants | Screen | Solubility | Activity | Mutants | Screen | Solubility | Crystal |
| F16 | F16A | | | N | | | | |
| | F16H | | | Y | | | | |
| | F16I | ✓ | ++ | | | | | |
| D23 | D23A | | | P | | | | |
| | D23N | ✓ | | | | | | |
| E26 | E26A | | ++ | N | | | | |
| | E26R | | +++ | N | E26R | | ++ | ✓ |
| | E26R/R64E | | | N | | | | |
| Y30 | Y30A | | | N | | | | |
| | Y30F | | | Y | | | | |
| | Y30H | | | Y | | | | |
| | Y30C | ✓ | + | | | | | |
| R37 | R37A | | | N | | | | |
| | R37K | ✓ | | | | | | |
| A44 | A44D | ✓ | | | | | | |
| | A44T | | ++ | | A44T | ✓ | ++ | ✓ |
| L48 | L48A | | | Y | | | | |
| | L48D | | | N | | | | |
| | L48K | | | N | | | | |
| R52 | R52A | | | P | | | | |
| D53 | D53A | | | N | | | | |
| W56 | W56A | | ++ | N | | | | |
| | W56F | | | Y | | | | |
| | W56Y | | | Y | | | | |
| | W56D | | | N | | | | |
| | W56E | | +++ | | | | | |
| | W56R | ✓ | ++ | | | | | |
| R64 | R64A | | | N | | | | |
| | R64E | | +++ | N | | | | |
| | R64Q | ✓ | ++ | | | | | |
| | R64C | | ++ | | R64C | ✓ | +++ | ✓ |
| | | | | | R64S | ✓ | +++ | ✓ |
| R65 | R65A | | | N | | | | |
| | R65E | | | N | | | | |
| | R65Q | ✓ | | | | | | |
| | | | | | R65S | ✓ | ++ | |
| | | | | | R65H | ✓ | | |

*Solubility is based on the estimated final yield of the purified protein per liter of bacterial culture. +++: >5 mg/L; ++: 1–5mg/L; +: <1 mg/L.
†Activity: N, no activity; P, partial activity; Y, activity close to wild-type.

## Probing the oligomerization interfaces of MAVS by a solubility screen

As an alternative verification of the key residues for MAVS CARD self-association, we searched for mutants that disrupt the CARD polymerization in *E. coli* (*Table 1*). We performed random mutagenesis and screened for soluble CARD mutants by using a recently developed solubility screen (*Harada et al.,*

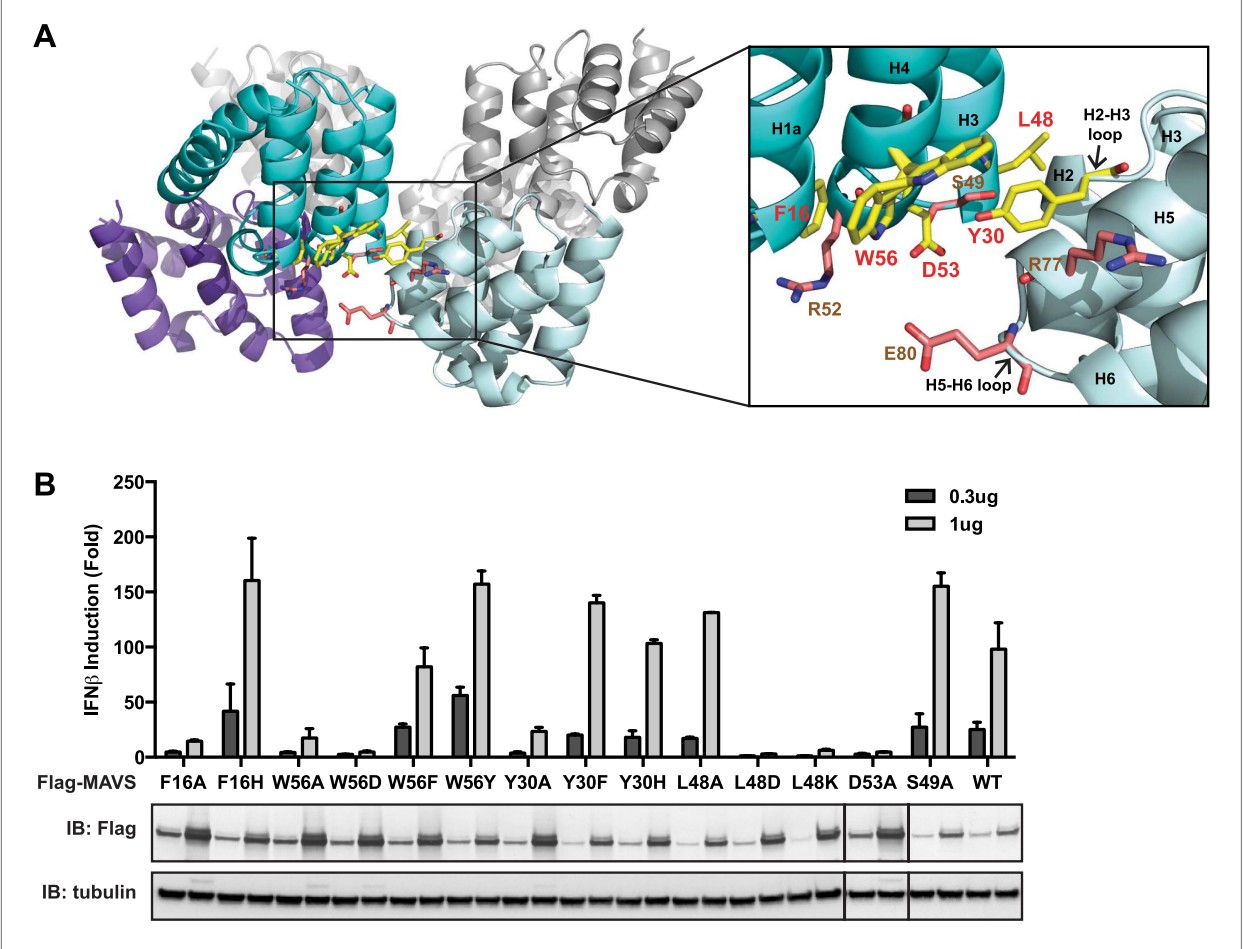

**Figure 3**. The intra-strand interface is mediated by hydrophobic interactions and hydrogen bonding. (**A**) MAVS CARD hexamer model with interacting residues at the intra-strand interface shown as yellow sticks. Residues that showed normal activity when substituted with alanine are displayed in brown. (**B**) MAVS proteins with point mutations at the intra-strand interface were tested for IFNβ induction and protein expression as in *Figure 2C*.

*2008*). Mutated CARDs fused to murine dihydrofolate reductase (mDHFR) were expressed. While mDHFR fused to wild-type CARD appeared in inclusion bodies due to CARD oligomerization, it remained soluble when it was fused a CARD mutant that failed to oligomerize. Expression of soluble mDHFR-CARD was selected by trimethoprim, which specifically inhibits bacterial DHFR, but not mDHFR. Our screen recognized multiple sites that are important for CARD oligomerization (*Table 1*; *Figure 2—figure supplement 1D*). Biochemical analysis confirmed that most mutants from the screen were expressed as soluble proteins, and eluted as homogenous monomers in gel-filtration chromatography (e.g., *Figure 2—figure supplement 1A*). Almost all the mutations out of the solubility screen are conserved and at the protein surface. They can be mapped to the two interfaces identified in the cryoEM model (*Figures 2 and 3*).

## CARD serves as the organization center for MAVS filaments

Previously we have shown that a deletion mutant of MAVS (MAVSΔProTM; *Figure 1A*) lacking part of the proline-rich region (PRR, residues 103–153) and the C-terminal transmembrane domain (TM; residues 461–540) allowed the production of a large amount of protein from *E. coli*, and remained capable of inducing wild-type MAVS to form fibers and potently activate IRF3 dimerization in cytosolic extracts (*Hou et al., 2011*). Compared to full-length MAVS, MAVSΔProTM is soluble when fused with SUMO, and can form functional polymers after removal of SUMO. Because MAVSΔProTM resembles the soluble part of the wild-type MAVS, its filament structure probably closely represents the polymerization of endogenous MAVS on mitochondria.

When we compared the EM images of the MAVSΔProTM filaments with those of the CARD filaments, the former apparently had extra mass and were larger in diameter than the latter (*Figure 4A*, *Figure 4—figure supplement 1*). We built a cryo dataset of 8909 boxed segments (particles) out of individual MAVSΔProTM filaments. The summed power spectrum of the raw particles exhibited

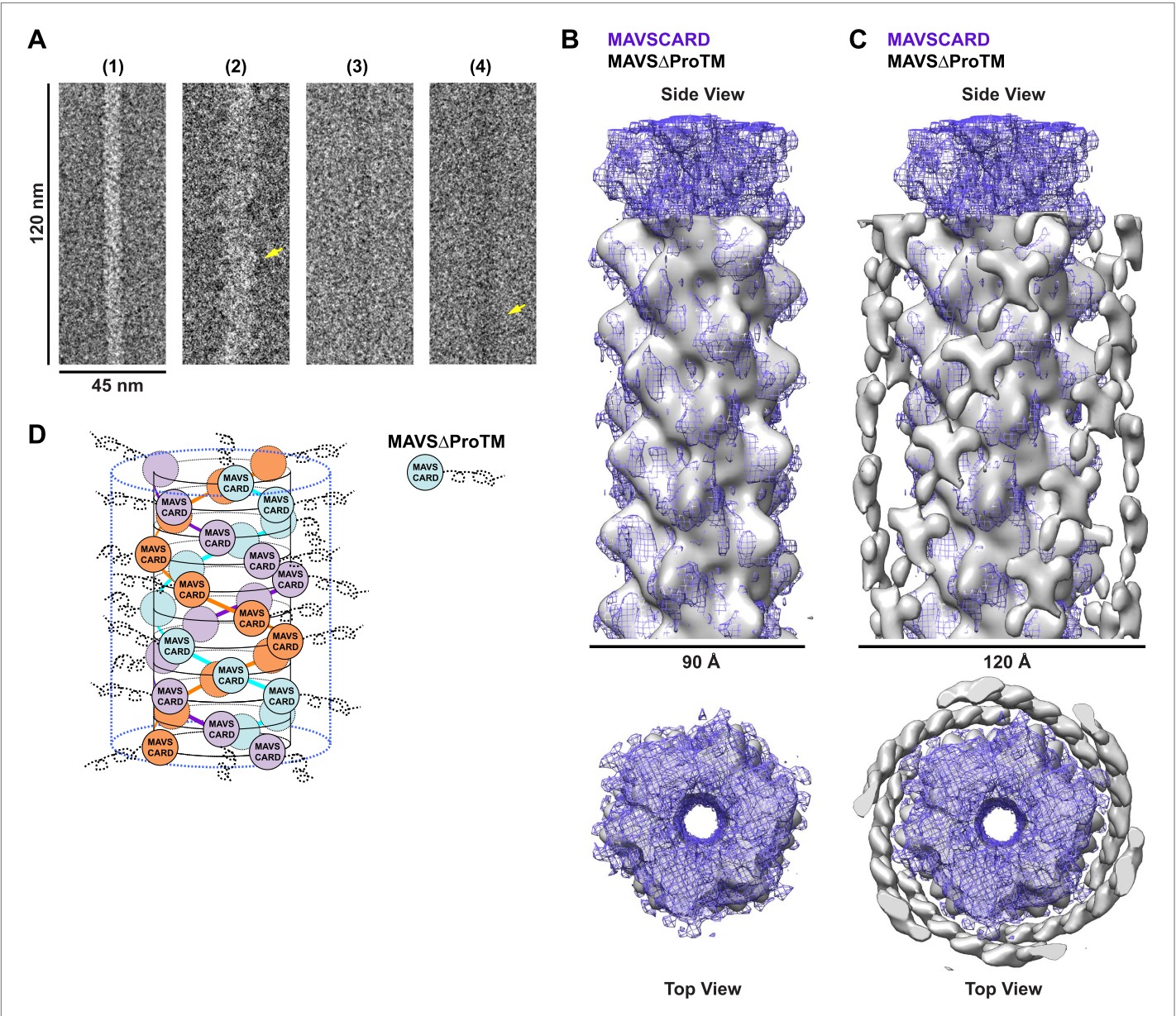

**Figure 4**. CARD filament is the organization center of the MAVS Filament. (**A**) Segments from negative-stain EM images of (1) Flag-MAVS CARD and (2) MAVSΔProTM and those from cryoEM images of (3) Flag-MAVS CARD and (4) MAVSΔProTM. For better visualization, protein is black in the cryoEM images. Yellow arrows point to the extra mass that made the MAVSΔProTM filaments larger in diameter. (**B**) Side and top views of the cryoEM reconstruction of MAVSΔProTM filament (gray, surface), which has the same helical symmetry as the MAVS CARD filament (purple, mesh). Only the middle 90 Å portion of the MAVSΔProTM map is shown. (**C**) Side and top views of the full MAVSΔProTM map (gray, surface) whose threshold was set at a proper level to overlap well with the CARD-only map (purple, mesh). A cylindrical sheet of extra density appeared at ~15 Å away from the central CARD filament. (**D**) Schematic view of the helical packing of MAVSΔProTM with the sequences C-terminal to the CARD shown as dashed coils.

The following figure supplements are available for figure 4:

**Figure supplement 1**. Image processing for the MAVSΔProTM filaments.

a similar pattern of layer lines as the CARD filaments (*Figure 4—figure supplement 1B* vs *Figure 1—figure supplement 1A*). We calculated a cryoEM map of MAVSΔProTM at a nominal 16.4 Å resolution (FSC$_{0.5}$; *Figure 4—figure supplement 1C*). Despite the extra mass, the MAVSΔProTM map is surprisingly similar to that of the CARD filament (*Figure 4B*). When a 90 Å-thick portion in the middle of the MAVSΔProTM map was compared with the map of the CARD filament (gray surface vs purple mesh in *Figure 4B*), the two overlapped very well at various threshold levels, suggesting that the core of the MAVSΔProTM fiber takes the same left-handed three-stranded helical structure. The symmetry parameters, a rotational angle of 52.9° and an axial rise of 16.9 Å, are almost the same as those of the CARD filament (*Figure 1C*).

When the whole MAVSΔProTM reconstruction is presented at a proper contour level so that the middle CARD filament has the same volume as the CARD-only filament, a cylindrical sheet of periodic densities appears at a distance of ~15 Å away from the perimeter of the central filament (*Figure 4C*, *Figure 4—figure supplement 1D,E*). MAVSΔProTM has ~300 extra residues C-terminal to its CARD. Some of these extra residues most likely contribute to the peripheral densities in the cryoEM map. These extra densities were clearly visible in the class averages calculated from the aligned filament images (red arrowheads in *Figure 4—figure supplement 1D*). Because of the intrinsic disorder in the segment between the CARD and the TM domain of MAVS, the majority of the extra residues did not produce significant density in the cryoEM map. A key conclusion from this comparison is that the CARD filament is in the center of the MAVS aggregates and the rest of the molecule is connected to the peripheral surface of the filament. Because the C-terminal TM domain is integrated in the mitochondrial membrane, the CARD filament and the membrane anchor the two ends of each MAVS molecule so that the intervening region is exposed to recruit cytosolic signaling effector molecules such as the TRAF (tumor necrosis factor receptor-associated factor) proteins (*Liu et al., 2013*; *Figure 4D*).

## CARD is crucial for the MAVS filament formation in cells

To visualize the filament formation of full-length MAVS, we stably reconstituted *Mavs-null* murine embryonic fibroblasts (MEFs) with Flag-tagged wild-type MAVS or its mutants. Although transient expression of wild-type MAVS resulted in constitutive signaling (*Kawai et al., 2005*; *Meylan et al., 2005*; *Seth et al., 2005*; *Xu et al., 2005*), the low expression level in the stable cell lines did not lead to constitutive activation of downstream target genes. Like endogenous protein, Flag-tagged MAVS was properly localized to the mitochondrial membranes, as demonstrated by its co-localization with either MitoTracker or TOM20, a 20 kDa subunit of the translocase in the outer mitochondrial membrane (*Figure 5—figure supplement 1, 3A*, Mock). Infection by Sendai virus induced the redistribution of MAVS protein and the formation of densely packed, speckled MAVS puncta on the surface of mitochondria, along with the nuclear translocation of NF-κB subunit p65 and induction of interferon-β (IFNβ; *Figure 5—figure supplement 1, 3*). In addition, Sendai virus also induced MAVS aggregation, which was detected by semi-denaturing detergent agarose electrophoresis (SDD-AGE; *Figure 5—figure supplement 3D*; *Hou et al., 2011*). Based on the nuclear translocation of p65 after viral infection, these bright MAVS puncta were observed in a majority of virus-infected cells (*Figure 5—figure supplement 3B*). In contrast, the bright puncta did not form in cells expressing MAVS mutants that failed to form filaments (*Figure 5—figure supplement 1, 2 and 3A*). As negative controls, mutants (E80A and F16H) that do not affect the MAVS filament formation were found to have normal puncta formation (*Figure 5—figure supplement 1, 2 and 3C*). Together these results support that the CARD-mediated aggregate formation is the key structural element for activating MAVS signaling in cells (*Figure 5—figure supplement 3C,D*).

Because the confocal fluorescent images of the MAVS aggregates did not have enough resolution to reveal the shape of the puncta in virus-infected cells, we next performed the experiments by Super-Resolution Structured Illumination Microscopy (SR-SIM) (*Gustafsson et al., 2008*). The resolution of conventional fluorescence microscopy is limited to ~200 nm in lateral (x, y) dimensions, and ~500 nm along the optical axis. SR-SIM increases both the lateral and axial resolutions by a factor of two (*Gustafsson et al., 2008*). SR-SIM images of Flag-tagged wild-type MAVS showed a fairly uniform distribution on the surface of mitochondria as it appeared in concentric rings around almost every MitoTracker-stained mitochondrion (*Figure 5A*, the last image in the top row). In cells infected with Sendai virus, the SR-SIM images revealed a clear redistribution of MAVS into rod-shaped clusters that interfaced with only a small fraction of mitochondria (e.g., white arrowheads in the rightmost image of the bottom row in *Figure 5A*). Because the expression level of MAVS in cells remained the same

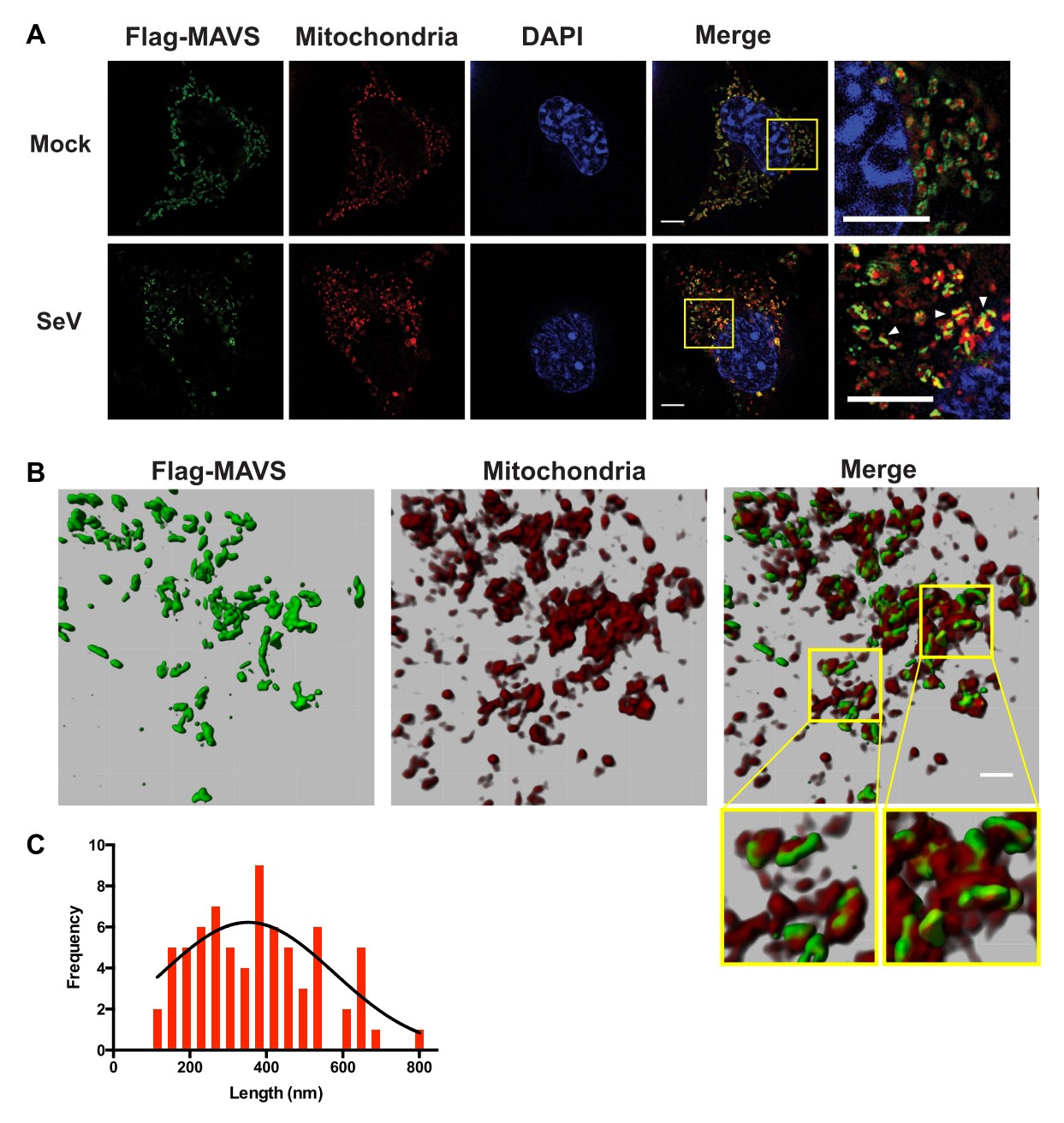

**Figure 5**. In virus-infected cells MAVS redistributes and forms rod-shaped puncta on the surface of mitochondria. (**A**) *Mavs*$^{-/-}$ MEF cells stably expressing Flag-tagged wild-type MAVS were mock-treated or infected with Sendai virus (SeV) for 12 hr and stained with MitoTracker (Mitochondria; red) and anti-Flag antibody (Flag-MAVS; green). Redistribution of MAVS among mitochondria was examined using SR-SIM. Expanded views of the areas within the yellow windows in the merged images were shown on the right. The SeV-infected cells contain bright foci of Flag-MAVS. The white arrowheads in the rightmost image of the bottom row highlight a few bright rod-shaped MAVS clusters. Scale bars, 5.0 μm. (**B**) 3D reconstruction of MAVS clusters (green) on the surface of mitochondria (red). Scale bar, 1.0 μm. The areas within the yellow windows in the merged image (right most) were expanded to show a few clusters that appear to bridge between mitochondrial membranes. (**C**) Histogram and Gaussian fit (black curve) of the length distribution that was measured from the SIM images of individual MAVS clusters as in panel **A** (SeV; N = 74) in virus-infected cells.

The following figure supplements are available for figure 5:

**Figure supplement 1**. Mutations at the inter-strand interface that disrupt CARD polymerization abolish the SeV-induced redistribution of MAVS on mitochondria.

*Figure 5. Continued on next page*

*Figure 5. Continued*

**Figure supplement 2**. Mutations at the intra-strand interface that disrupt CARD polymerization abolish the SeV-induced redistribution of MAVS on mitochondria.

**Figure supplement 3**. Strong correlations among MAVS puncta formation, MAVS signaling and MAVS CARD polymerization.

before and after viral infection (*Figure 5—figure supplement 3D* and references) (*Hou et al., 2011*; *Liu et al., 2013*), the redistribution of MAVS from one mitochondrion to another during the puncta formation may result from mitochondrial fusion (*Yasukawa et al., 2009*; *Castanier et al., 2010*; *Koshiba et al., 2011*). Alternatively, MAVS aggregates on some mitochondria may be degraded through an unknown mechanism, which is less likely because of the unchanged level of MAVS protein. The diameter of the rod-shaped MAVS clusters is probably less than 100 nm, the lateral resolution limit of SR-SIM.

When a series of SR-SIM images were combined in IMARIS to reconstruct a 3D volume of MAVS and mitochondria in virus-infected cells (*Figure 5B*), it became clear that the MAVS aggregates were not evenly distributed around individual mitochondria, but were clustered into narrow regions on the surface of mitochondria (the rightmost image in *Figure 5B*). Many of these rod-shaped clusters bridge between two or more mitochondrial membranes, suggesting that MAVS molecules from multiple mitochondria may contribute to the formation of one MAVS filament (magnified insets of *Figure 5B*).

To quantify the average number of MAVS molecules for the rod-shaped clusters, we measured the length distribution of many observed clusters in the 2D SR-SIM images (*Figure 5A,C*). The mean filament length in the observed Poisson distribution is approximately 400 nm (full width at half maximum, or FWHM, n = 74) and the longer ones are ~800 nm. Based on the cryoEM map, each 400 nm filament contains approximately 720 MAVS molecules.

## Discussion

The pseudoatomic model for the MAVS CARD filaments from our cryoEM study suggests that after viral infection, activated MAVS molecules on the mitochondrial surface interact with each other at both the intra- and inter-strand interfaces between their CARD domains (*Figure 6*). The MAVS aggregates in cells are indeed rod-shaped clusters that may contain MAVS molecules from multiple mitochondria. The CARD filaments form the central elements of the MAVS aggregates, and can promote their own growth by attracting new CARDs into the pre-poised interaction interfaces. The filaments are localized on the mitochondrial surface and the MAVS TM domains are embedded inside the outer mitochondrial membranes. These two ends of each MAVS molecule provide important spatial constraints that may force the intervening coiled sequence to be well extended and exposed for recruiting down-stream signaling molecules (*Liu et al., 2013*; *Figure 6*). Bioinformatic analysis suggests that the middle segment of MAVS forms random coils, which, if present by themselves in aqueous phase, would not likely be fully extended and thus may deter efficient binding of multiple positive or negative regulators of MAVS. The spatial arrangement between the CARD filament and the mitochondrial membrane provides a good solution to this problem of intrinsic disorder. The prion-like filament formation of MAVS thus uses very different chemistry than other prion proteins, and orchestrates the signaling domains of MAVS into a high-affinity platform for rapid and efficient signaling.

The filament formation of MAVS CARDs is based on collective interactions at the four interfaces of each subunit (*Figures 1D, 2B and 3A*). These interfaces maintain the tight and dense packing of individual CARDs in the filaments, which, together with the strong inter-strand electrostatic interactions, probably make the filaments detergent-resistant. During filament formation, there are counter-acting forces, such as decrease in entropy and possible structural adjustment at the pore-lining surface of the CARD filament (*Figure 1C*; *Video 2*), which would increase the energy level of the filaments. The net decrease in free energy supporting the filament formation is due to the four interaction interfaces for every CARD. But the counter-balance of these positive and negative energy terms likely makes the filaments fairly sensitive to mutations. Indeed, the four interfaces appear to be dominated by hot spots because point mutations in multiple positions are capable of destabilizing the filaments. The open ends of the CARD filaments provide three binding sites for each incoming CARD molecule. Because of the net gain in free energy for each CARD, the longer filaments are expected to be more stable than

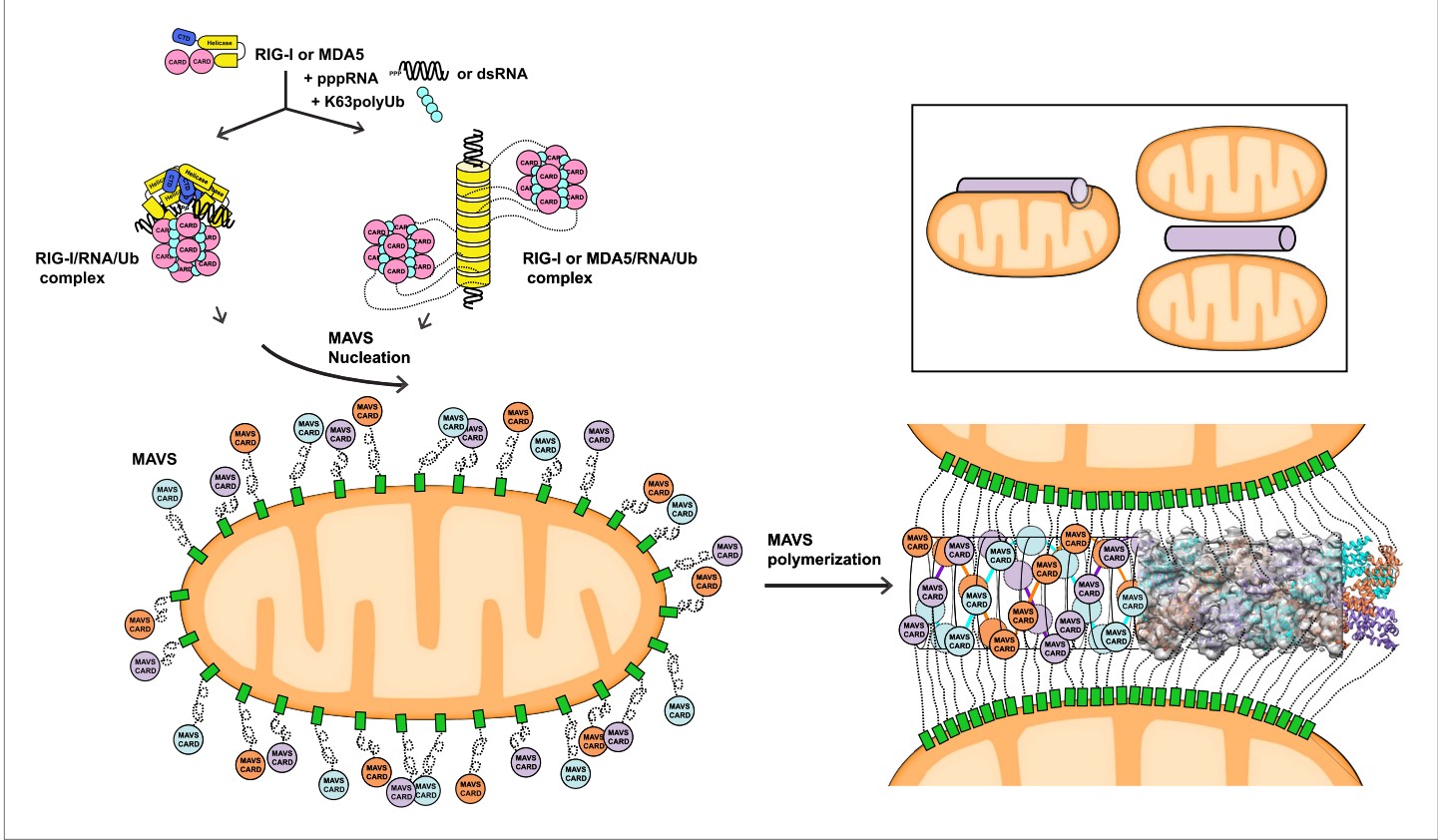

**Figure 6**. A working model for MAVS activation by RIG-I/RNA complexes. Detection of 5'-pppRNA by RIG-I or dsRNA by MDA5 or RIG-I triggers the formation of RIG-I (or MDA5)/RNA/polyUb complex. The CARD domains of individual complexes are poised properly to attract three MAVS CARD domains and support the nucleation of the filament. In the resting state, MAVS CARD is sequestered and has a low probability of forming polymers. The RIG-I (or MDA5)/RNA/polyUb complexes stabilize the MAVS CARDs in the exposed state and bring three copies together to initiate the filament formation. Once started, a short MAVS CARD filament promotes its own elongation by attracting more MAVS CARDs into the assembly. The filament can form on the surface of one mitochondrion or between two or more mitochondrial membranes. Inset, one or more mitochondria might be involved in MAVS filament formation.

the shorter ones, leading to the filaments that are up to 10 microns long under cryoEM conditions. Inside virus-infected cells, the average length of the rod-shaped MAVS clusters is merely ~400 nm. The physical length of MAVS filaments in cells may be limited by the number of MAVS molecules in each mitochondrion and by the dynamics of mitochondrial fusion and fission (*Onoguchi et al., 2010*; *Figure 5C*).

The three-stranded helical filaments at the center of the MAVS aggregates and the complete separation of three CARD units in each layer suggest that a successful initiation of filament formation needs at least two independent molecules (*Figure 6*). These independent molecules need to be juxtaposed in order to poise three CARDs to form the first layer of the filament. The tandem CARDs in one RIG-I molecule are restrained by a short α-helix running between them, and are less likely to serve this role. There are two possible scenarios for two RIG-I/RNA complexes to join for initiation: (1) Two individual RIG-I/RNA monomers can come together; or (2) RIG-I molecules may form dimers or oligomers along short duplex RNAs. Crystallographic studies of RIG-I (full-length or truncated) have led to the proposal that the RIG-I/RNA complex might act as monomers in conjunction with polyubiquitin to activate MAVS (*Civril et al., 2011*; *Hild et al., 2011*; *Jiang et al., 2011*; *Kowalinski et al., 2011*; *Lee et al., 2011*; *Luo et al., 2012, 2013*). Should two or more RIG-I/short RNA monomeric complexes be able to nucleate a MAVS filament, we would expect the short RNAs (10–17 bp) used in the crystallographic studies to activate the pathway. But the published data did not lead to a consensus on the activities of these short RNAs (*Fujita, 2009*; *Schlee et al., 2009*; *Marq et al., 2011*; *Kohlway et al., 2013*). Recent biochemical data showed that RIG-I tandem CARDs and K63 polyubiquitin form relatively stable 4:4 oligomers, which may nucleate the helical filamentous structure formed by the MAVS CARDs (*Figure 6*; *Jiang et al., 2012*). In a similar fashion, the RIG-I (or MDA5)/RNA helical complex

may present multiple CARD domains on its periphery where multiple MAVS filaments may be initiated (*Berke and Modis, 2012*; *Wu et al., 2013*). Moreover, negative-stain EM studies have suggested either parallel or anti-parallel dimers to be the active form of RLRs (*Murali et al., 2008*; *Ranjith-Kumar et al., 2009*). Structural studies of active RIG-I-RNA and RIG-I-MAVS complexes are needed to provide a detailed mechanism by which RIG-I initiates MAVS polymerization.

Protein oligomerization mediated by the death domain (DD) superfamily, which include CARDs, has been studied in multiple signaling proteins. For example, the helical assembly of MyDDosome was proposed to use three different types of DD–DD interfaces (*Lin et al., 2010*; *Gay et al., 2011*; *Kersse et al., 2011*), none of which share any similarity to the intra- or inter-strand interfaces in the MAVS CARD filament (*Figure 1D*). The MyDDosome formation is thought to bring multiple kinase domains of the IRAKs together for efficient activation. This affinity-enhancing scheme for recruiting downstream effectors is essentially the same as in the MAVS filaments. The much longer MAVS filaments would thus entail a significant amplification of the signal that nucleates their formation and enhance the RLR sensitivity to viral RNAs. Because TRAF2 and TRAF3 both form trimers (*Park et al., 1999*; *Ni et al., 2000*; *Zheng et al., 2010*; *Napetschnig and Wu, 2013*), MAVS filaments may enhance the TRAF-MAVS interaction through increased avidity, and boost the signaling efficacy significantly.

The redistribution of MAVS molecules on the surface of mitochondria and the possible bridging of multiple mitochondria by the MAVS filaments (*Figures 5, 6*) were apparently a result of the filament formation. The filament growth needs to bring randomly distributed MAVS into close vicinity. The limited number of MAVS molecules per mitochondrion suggests that the long MAVS filaments may be contributed by multiple mitochondria. The clustered MAVS molecules become sequestered during the dynamic mitochondrial fission and fusion cycles. The fusion of sequestered clusters leads to the redistribution of MAVS from being ubiquitously present in almost all mitochondria to being segregated on a small number of them. The self-promoting nature of in vivo MAVS filaments may be the driving force that leads to almost all MAVS being incorporated into filaments (*Figure 5C*, *Figure 6*). Further analyses of MAVS redistribution and its correlation with mitochondrial fusion/fission dynamics by live-cell imaging and computational analysis may provide more detailed insights into this process.

In summary, our structural and functional studies of the MAVS polymerization revealed the unique three-stranded filaments and identified the interaction interfaces that drive the formation of self-perpetuating prion-like fibers. The structural organization of the MAVS filaments on mitochondrial membranes is distinct from other oligomeric assemblies such as the MyDDosome and β-amyloid prions. This virtually irreversible polymerization of MAVS provides a highly sensitive and robust mechanism of immune response. Such digital (i.e., all or none) response allows the organisms to defend against noxious agents such as lethal infections.

## Materials and methods

### Reagents and standard methods

Mouse antibody against Flag-tag (M2) and M2-conjugated agarose were purchased from Sigma-Aldrich (St. Louis, MO); rabbit antibodies against TOM20 and the p65 subunit of NF-κB were from Santa Cruz Biotechnology (Dallas, TX); Alexa Fluor 488 conjugated goat anti-mouse and anti-rabbit antibodies, Alexa Fluor 568 conjugated goat anti-mouse antibody, and Alexa Fluor 633 conjugated goat anti-rabbit antibody were from Invitrogen (Carlsbad, CA). Sendai virus (SeV, Cantell strain, Charles River Laboratories) was used at 100 hemagglutination (HA) units/ml culture media. HEK293T, HEK293T-IFNβ-luciferase, *Mavs*$^{-/-}$ MEF cells and derivatives were cultured in Dulbecco's modified Eagle's medium (DMEM) supplemented with 10% (v/v) cosmic calf serum (Hyclone, Thermo Fisher Scientific, Waltham, MA) with penicillin (100 U/ml) and streptomycin (100 μg/ml). Other chemicals and reagents were from Sigma-Aldrich unless otherwise specified.

### Protein expression and purification

cDNA encoding Flag-tagged MAVS CARD (1–100) has been described previously (*Hou et al., 2011*). For protein expression, pcDNA3-Flag-MAVS CARD was transiently transfected into HEK293T cells. Cells were harvested 36 hr after transfection and lysed in a buffer containing 20 mM Tris–HCl (pH 8.0), 150 mM NaCl, 10% glycerol, 0.10% Triton X-100, 1.0 mM DTT, and EDTA-free protease inhibitor cocktail (Roche, Basel, Switzerland). After centrifugation at 10,000×*g* for 10 min, Flag-MAVS CARD was selectively bound to M2-antibody-conjugated agarose beads and eluted by Flag peptide. The eluate was fractionated on a

Superdex 200 PC 3.2/30 column (GE Healthcare, Uppsala, Sweden) equilibrated in a buffer containing 20 mM Tris–HCl (pH 7.5), 50 mM NaCl and 1.0 mM DTT. Fractions were analyzed by SDS-PAGE and silver staining.

cDNA encoding the MAVSΔProTM mutant lacking the proline-rich region (103–153) and the C terminal transmembrane domain (461–540) has been described previously (*Hou et al., 2011*). The bacterial expression vector pET-28a-His$_6$-Sumo-MAVSΔProTM was transformed into BL21 (pLys). Protein expression was induced with 0.20 mM IPTG at 18°C for four hours. After sonication in a lysis buffer containing 10 mM Tris–HCl (pH 8.0), 500 mM NaCl, 0.50 mM DTT, 5.0% glycerol, 0.50 mM PMSF and 10 mM imidazole, cell lysates were centrifuged at 50,000×*g* for 30 min. His$_6$-Sumo-MAVSΔProTM in the supernatant was purified using Ni-NTA affinity resin (QIAGEN, Limburg, Netherlands). Subsequently, the protein was loaded onto HiTrap Q HP column (GE Healthcare), and then eluted with a gradient of NaCl varying from 0.10 M to 0.50 M in a buffer made of 10 mM Tris–HCl (pH 7.5), 5.0% glycerol, 2.0 mM DTT, 1.0 mM EDTA and 0.50 mM PMSF. The fractions containing His$_6$-Sumo-MAVSΔProTM, which were eluted with 300 mM NaCl, were pooled together and applied to a Superdex 200 HR 10/30 column (GE Healthcare) equilibrated with a buffer made of 10 mM Tris–HCl (pH 8.0), 150 mM NaCl, 1.0 mM DTT, 1.0 mM EDTA and 0.50 mM PMSF. His$_6$-Sumo-MAVSΔProTM was then digested with SUMO protease at 4.0°C overnight. The His$_6$-SUMO tag was removed by running the reaction mixture in a Superdex 200 PC 3.2/30 column (GE Healthcare), which was equilibrated in a buffer containing 10 mM Tris–HCl (pH 8.0), 150 mM NaCl and 1.0 mM DTT. The peak fraction of the protein was collected for EM studies.

## Negative-stain electron microscopy

Copper grids (Ted Pella Inc., Redding, CA) coated with a layer of thin carbon film (3–5 nm) were rendered hydrophilic by negative glow discharge in air. A 2-4 µl aliquot of the purified MAVS sample was loaded onto the grids. After 30 s of incubation on the grid at room temperature, the sample was stained with 2.0% phosphotungstic acid (PTA) at pH 8.0 and blotted dry. Samples were imaged in a JEOL 2200FS FEG electron microscope operated at 200 kV with a nominal magnification of 50,000 × (2.84 Å/pixel at the detector level) using a defocus range of −0.7 to −1.5 µm. Images were recorded with an electron dose of 20 e$^-$/Å$^2$ on a 2K × 2K Tietz slowscan Charge Coupled Device (CCD) camera.

## CryoEM sample preparation and data collection

Quantifoil R2/2 grids (Quantifoil Micro Tools GmbH, Jena, Germany) were coated with a thin carbon film (1–3 nm) in order to retain more filaments for imaging. Right before use, the grids were negatively glow-discharged in air. 2.5 µl purified MAVS was loaded onto the grids. Grids were blotted in 100% humidity at 4°C for 5 s before being plunge-frozen into liquid ethane bathed in liquid nitrogen inside a Vitrobot (FEI, Hillsboro, OR). After the specimens were transferred into and kept frozen inside the JEOL 2200FS FEG electron microscope, images were recorded on SO163 films (Eastman Kodak, Rochester, NY) with a nominal magnification of 60,000 × under low-dose conditions (~20 e$^-$/Å$^2$). A Gatan K2 Summit Direct Detector (Gatan, Pleasanton, CA) was used for testing specimens in the later phase of the project. A total of 358 films were developed using full-strength D19 (Kodak) solution. Micrographs were digitized with a PhotoScan film Scanner (Z/I Imaging GmbH, Germany) at a step size of 7.0 µm. After 2 × 2 binning, the pixel size was 2.33 Å on the specimen. The magnification calibration at the nominal 60,000 × was 61,950 × and the actual pixel size was 2.26 Å. Datasets from other imaging conditions were scaled to this condition after the axial rise of each dataset was independently determined through IHRSR. A total of 30,384 segments were boxed from film data using the EMAN2 program HelixBoxer (*Ludtke et al., 1999*).

To accelerate data collection, we sent the grids of the MAVS CARD filaments to the HHMI Janelia Farm Research Campus, and collected ~2,100 images with a 4K × 4K Falcon Direct Detector in a Titan Krios microscope. The microscope was operated at 300 kV and was equipped with a *Cs* corrector. Automatic data collection was run by proprietary software, EPU (FEI, Hillsboro, OR). Images were taken under −2.5 to −4.0 microns of defocus at 29,000 ×, which gave rise to a calibrated pixel size of 2.30 Å at the specimen level. The density of filaments in the cryo specimens was fairly low so that we had to collect a large number of images. After the evaluation of their power spectra, 1,088 Falcon images were selected, and a total of 18,500 short filaments were boxed out for analysis.

## EM image processing

The analysis of the datasets of both the MAVS CARD filaments and the MAVSΔProTM filaments followed the IHRSR method developed by Dr Edward Egelman's group at University of Virginia. The method was implemented in SPIDER. Dr Egelman kindly provided the programs in SPIDER, and

the HSEARCH_LORENTZ/HIMPOSE programs. All SPIDER programs were rewritten in order to run with SPIDER Version 19.08 in a Linux cluster that runs the Redhat Enterprise 5.0. Besides, new SPIDER programs were written to refine the map for out-of-plane tilting of the filaments and for local optimization in angle assignment. Extensive technical details could be found in reference (*Mukherjee et al., 2014*). Briefly, from all images obtained in one session of data collection, the filaments were boxed out into 200 pixel segments (~45.2 nm long at the specimen level) with 90% overlapping between neighboring ones from the same long filament. Filaments shorter than 200 pixels were discarded. The filament helical axis was always positioned roughly along the Y-axis. The defocus and astigmatism information for each image were determined by CTFFIND2 in the MRC package (IMAGE2010). The filament particles (boxed segments out of the raw filaments) were phase-flipped and band-pass filtered. We also tested the Wiener-filtering with a constant of 0.2 (1/SNR), and did not find significant difference from the phase-flipped dataset. Afterwards the filaments from different images out of one cryo session were pooled together as a subdataset. To start the analysis, we made sure that the summed power spectrum of each subdataset showed a faint (usually fuzzy) layer line at the position of the first meridional line (layer line 9; *Figure 1—figure supplement 1A*, *Figure 4—figure supplement 1B*). The diameter of individual filaments was estimated from the raw images. A cylindrical volume that had the estimated diameter was generated in SPIDER and projected along Y-axis to be used as the first reference. The individual filaments were compared with the projection from the cylindrical volume. The shift in X-direction for each boxed segment (particle) was rounded to the nearest integer and applied to center the particle horizontally. The total power spectrum was calculated from all X-centered particles in the dataset. The four layer lines (LL = 1, 4, 5, and 9) were measured to prepare a (LL, Z*) table. The peak positions were indexed to make a table of (n, R). The n was estimated from a numeric table of Bessel functions from 0 to 14$^{th}$ orders. The helical indexing led to the initial guess of a 3-start helix (n = 3 or −3), with some uncertainty of a 4-start one (n = 4 or −4). We then tested the refinement for both a 3-start helix and 4-start one separately, and found that the model with a 3-start helical symmetry was able to converge to a 3D map with the right axial rise, but not the one with a 4-start symmetry (*Figure 1—figure supplement 1C,D,G*). Full refinement with n = −4 was done to confirm this point. We therefore concluded that n = 3 was probably right (n = −3 is the mirror and equally possible). With this information, we did the initial analysis of all subdatasets from different cryo sessions independently in order to obtain the rotation angle (ΔΦ) and the axial rise (Δz) for each subdataset. The axial rises for individual datasets collected at different magnifications or from different microscopes were brought together and compared to find the right parameters for interpolating the images in the different datasets and scaled them all together into one large dataset that contained 48,884 particles. The refinement of the large dataset against a cylindrical volume of a diameter of 90 Å quickly led to a stable solution (*Figure 1—figure supplement 1C,D*) with the symmetry parameters of ΔΦ = 53.6° and Δz = 16.5 Å. To test the robustness of the parameters, we also introduced 1.0 Å deviation to the axial rise in the middle of the refinement, and observed the quick recovery of the refinement to the stable solution (*Figure 1—figure supplement 1D*). This test suggested that the solution was quite stable.

When we performed multivariate statistical analysis and hierarchical classification of the large dataset in IMAGIC, the Eigen images clearly showed that local bending and symmetry breakdown (*Figure 1—figure supplement 1B*) appeared to be the major defects in the dataset, which might have been a limiting factor to a better resolution. We tried to eliminate a small fraction of images (~20%) that corresponded to those class-averaged images showing clearly weak symmetry or distortions. But after that, we still saw distortions in the Eigen images calculated from the rest of the images. We therefore did not eliminate ~50% or more of the data to examine the resolution. After an initial refinement, we sorted the data into nine bins by aligning them against models whose symmetry parameters are centered around (ΔΦ = 53.6°, Δz = 16.5 Å) with ΔΔΦ = 2.0° and ΔΔz = 1.5 Å. The distribution of the particles into these 9 bins followed a Gaussian distribution, and the central bin had 20,825 particles and was selected for further refinement where the out-of-plane tilting was refined in 2° steps to a maximum of 15°. After 140 runs the final map was stable and the symmetry well converged to ΔΦ = 53.6° and Δz = 16.8 Å. The final map was calculated from 15,366 boxed segments. The map after IHRSR was corrected by an average CTF envelope calculated from the CTF parameters for all images used in the final map calculation. After the FSC calculation (see next paragraph), a negative B-factor of −1,100 Å$^2$ was applied to sharpen the final map to 9.6 Å.

To estimate the resolution by FSC, the sorted dataset (20,825 particles) was separated into top and bottom two halves. Two completely independent volumes were obtained by refining the two half datasets against a cylinder. The two refinements starting with different initial symmetry parameters

converged to almost exactly the same symmetry parameters: $\Delta\Phi = 53.61°$ and $\Delta z = 16.68$ Å for the first, and $\Delta\Phi = 53.66°$ and $\Delta z = 16.66$ Å for the second. The two volumes without symmetry imposition were aligned in UCSF Chimera (*Pettersen et al., 2004*). The second volume was shifted by 0.225 pixels and rotated by −28.96° around Y-axis before the FSC calculation (*Figure 1—figure supplement 1E*). The estimated resolution at FSC = 0.5 was 9.6 Å. Because of almost exactly the same symmetry in two volumes, we worried about noise correlation and did not use the 0.143 threshold for our resolution estimate. As a control, we calculated the FSC before the two volumes were aligned, and found that the FSC quickly fell down to below 0.1 at around 40 Å. To confirm that the final map recapitulates the characteristic of the original dataset, we calculated the summed power spectrum of projections from the map at 360 different orientations around its helical axis (*Figure 1—figure supplement 1A*, right). The power spectrum showed the same patterns as in the total power of the raw dataset (*Figure 1—figure supplement 1A*, left), with layer lines extending to higher resolution range.

Analysis of the MAVSΔProTM dataset followed the same procedure. The dataset had only 8,909 particles (*Figure 4—figure supplement 1A,D,E*). We therefore did not sort them against models with varying symmetry parameters. The FSC calculation gave rise to the resolution estimate of 16.4 Å (*Figure 4—figure supplement 1C*).

## Cryo-electron tomography (cryo-ET)

Energy-filtered electron cryo-tomography on the MAVS CARD filaments was carried out from a dataset collected with an FEI Titan Krios cryo-EM at HHMI Janelia Farm Research Campus. The tomography tilt series were collected in a Gatan K2 Summit direct electron detector installed behind a GIF Quantum energy filter. A narrow energy slit of 5 eV was used together with a small objective aperture (50 microns) to enhance image contrast. The data were collected at a nominal magnification of 42,000 ×, corresponding to 2.7 Å per pixel on the K2 camera. The tilt range spans from −60 to +60° with a step size of 3°. The defocus level was set at −6.0 microns. Tomographic reconstruction was carried out using the standard weighted back projection procedure implemented in IMOD (*Yu et al., 2013*). The tilt series were aligned using patch tracking due to the absence of fiducial gold particles. A non-linear anisotropic diffusion filter was applied to the reconstruction. For better resolution of the helical stripes, five slices that cover the top surfaces of multiple filaments in the tomogram were used to calculate a projection image (*Figure 1—figure supplement 2B*), and the view was from above the filaments. The boxed segments were cropped out and the distance between stripes was measured in the image. The angle between the stripes and the helical axis was estimated manually.

## Docking the MAVS CARD crystal structure into the EM map

The crystal structure of MAVS CARD (PDB Code: 2VGQ) was first docked into the cryoEM density map manually in UCSF Chimera. We tested the docking in the original map and its mirror, and found that the docking to the mirrored map, which has left-handed symmetry, allowed us to position three α-helices of the X-ray model directly into the density features in the map (*Figure 1C*). Recognition of these surface features led to a fairly unique position for the X-ray model and resolved the ambiguity in chirality, consistent with the cryoET results. After one CARD was docked into the map, a hexamer model was built by applying symmetry operations. We segmented the cryoEM density and extracted a portion that represents a hexamer formed by two layers of protein subunits using the segmentation tools in UCSF Chimera. Subsequently, the hexamer model was optimized using SITUS (*Wriggers, 2010*) to find a local best position with the symmetry constraints. The side chains of a few charged residues (D23, E26, R37, R64 and R65) at the inter-strand interface in the model were optimized by testing different rotamers in Coot to produce the model in *Figure 2B* (*Emsley et al., 2010*).

The two videos were made with Chimera.

## Screen for soluble MAVS CARD mutants in *E. coli*

Creation and solubility screen of mutant libraries of the human and horse CARDs were performed as described previously (*Harada et al., 2008*). Briefly, the coding regions of the MAVS CARD (residues 2–98) were randomly mutagenized by using error-prone PCR, and sub-cloned into the pBAD/DHFR vector (kindly provided by Dr James Bowie at UCLA) between the NcoI/SalI restriction sites. Plasmids were transformed into Top10 cells (Invitrogen) to express CARD mutants fused to the C-terminus of DHFR. Cells were grown on M9 minimal medium plates containing 100 μg/ml ampicillin, 0.20% arabinose and with or without 1 μg/ml trimethoprim at 37°C for 72 hr. Colonies that grew in the presence of trimethoprim

were candidate clones expressing soluble CARD mutants. Plasmids from these clones were sequenced to identify their mutations in the CARD domain. The mutants were subsequently subcloned into a modified pET28a vector that encodes an N-terminal His$_6$-tag and a cleavage site of human rhinovirus C3 protease (*He et al., 2009*), and expressed in the bacterial strain BL21 DE3 using a standard procedure. Proteins were purified using Ni-NTA and gel filtration chromatography. The N-terminal tag was removed by treatment of human rhinovirus C3 protease. Mutants were identified as soluble if the proteins could be purified and behaved as a homogenous monomeric peak in gel filtration chromatography.

## Crystallization and structure determination of soluble MAVS CARD mutants

Well-behaving CARD mutants were concentrated and subjected to crystallization trials. Many mutants of horse CARD crystallized in multiple conditions, many of which belonged to the same crystal form as shown by preliminary diffraction experiments. The E26R and R64C mutants, both mapped to the electrostatic interface for mediating MAVS oligomerization, were chosen for structure determination. Crystals of E26R at 1.1 mg/ml grew in 0.10 M MES (pH6.5), 30%PEG 5000 MME, 0.20 M ammonium sulfate at 20°C. Crystals of R64C at 1.5 mg/ml grew in Bis-Tris (pH6.5), 25% PEG 3350, 0.20 M ammonium acetate at 20°C. Crystals were flash-cooled in liquid nitrogen in their crystallization buffers supplemented with 25% glycerol. Diffraction data for E26R were collected at 100 K in Beamline 19ID of the Advanced Photon Source (Argonne National Laboratory). Data for R64C were collected at 100 K with a Rigaku X-ray source using a Raxis IV detector. Diffraction data were indexed, integrated and scaled by using HKL2000 (*Otwinowski and Minor, 1997*). The structure of the human MAVS CARD (PDB ID: 2VGQ) was used as the search model for molecular replacement using Phaser in the Phenix package (*Adams et al., 2002*; *McCoy et al., 2007*). Iterative model building and refinement were performed in Phenix and Coot, respectively (*Emsley and Cowtan, 2004*). The data collection and refinement statistics were summarized in *Supplementary file 1*.

## Interferon-β luciferase reporter assay

HEK293T cells stably expressing both Renilla luciferase (as an internal control) and IFNβ promoter driving firefly luciferase were transfected with the indicated amounts of cDNAs for Flag-tagged wide-type MAVS or its mutants. 24 hr after transfection, cells were harvested to measure the expression of luciferase using a dual luciferase assay kit (Promega, Madison, WI).

## Confocal and SR-SIM imaging

*Mavs*$^{-/-}$ MEFs stably expressing wide-type MAVS or its mutants were grown on sterile glass coverslips in 12-well plates. 12 hr after Sendai virus infection, cells were first stained with MitoTracker Red according to the manufacturer's instructions (Invitrogen). Cells were then fixed with 4.0% paraformaldehyde in PBS for 15 min, permeabilized in PBS containing 0.10% Triton X-100 for 5 min, and blocked in PBS containing 0.10% Triton X-100 and 10% BSA for 30 min at room temperature. After blocking, the cells were incubated with specific primary antibodies for 1 hr, washed and then incubated with suitable Alexa Fluor 488 (or Alexa Fluor 568 or Alexa Fluor 633)-conjugated secondary antibodies for another hour. After careful wash, the slides were mounted with the VECTASHIELD mounting medium with DAPI (Vector Laboratories). Imaging of the cells was carried out using a Zeiss LSM510 META laser scanning Confocal Microscope or a Zeiss ELYRA PS.1 Super-Resolution Structured Illumination Microscope (SR-SIM). Z-stacks with an interval of 110 nm were used to section the whole cell for 3D-SR-SIM. Images were analyzed using the Zen2011 software (Zeiss) or ImageJ. Alignment and reconstruction of 3D-SIM images were performed using IMARIS (Bitplane).

## Quantitative reverse transcription PCR (q-RT-PCR)

Total RNA was isolated using TRIzol (Invitrogen). 0.1 μg of total RNA was reverse-transcribed into cDNA with iScript cDNA synthesis kit (Bio-Rad, Hercules, CA). The resulting cDNAs served as the templates for Quantitative-PCR analysis using iTaq Universal SYBR Green Supermix (Bio-Rad) and ViiTM7 Real-Time PCR System (Applied Biosystems Inc., Foster City, CA). Primers for specific genes are: Mouse β-actin, 5′-TGACGTTGACATCCGTAAAGACC-3′ and 5′-AAGGGTGTAAAACGCAGCTCA-3′; Mouse IFNβ, 5′-CCCTATGGAGATGACGGAGA-3′ and 5′-CTGTCTGCTGGTGGAGTTCA-3′.

## Semi-denaturing detergent agarose gel electrophoresis (SDD-AGE)

The formation of prion-like aggregates of MAVS and its mutants was analyzed by SDD-AGE as previously described (*Hou et al., 2011*).

## Acknowledgements

The authors would like to acknowledge the assistance of the UT Southwestern Live Cell Imaging Facility, a Shared Resource of the Harold C Simmons Cancer Center under the supervision of Dr Kate Phelps and supported in part by an NCI Cancer Center Support Grant, 1P30 CA142543-01. We are grateful to Dr Bryant Chhun at Carl Zeiss Microscopy, LLC for his technical expertise in helping us collect the SR-SIM images, Dr Yue Ma (UT Southwestern Medical Center) for her help with the cell sorting experiments, Ms Hima Lingam for particle selection, and Dr Kate Phelps for her insightful evaluation of the SIM images and her assistance in image interpretation. We thank Drs Steven Altschuler and Lani Wu and Mr Adam Coster (UT Southwestern Medical Center) for advice on the statistical analysis of fluorescent images of cells, Ms Robyn Roth (Washington University at St Louis) for freeze-etch EM, Dr James Bowie at UCLA for providing the plasmid for the solubility screen and Dr Edward Egelman at University of Virginia for sharing the software package for IHRSR and for his advice on helical indexing and helical reconstruction. We are grateful to other members of the Jiang, Chen and Zhang laboratories for various supports. This work was supported by grants from NIH (R01 GM088745 and GM093271 to Q-XJ; R01 AI093967 to ZJC), and Welch Foundation (I-1684 to Q-XJ; I-1389 to ZJC). Dr Jiang is also supported in part by an AHA National Innovative Award (12IRG9400019) and a CPRIT grant (RP120474). The K2 Summit Direct Detector recently introduced to the cryoEM at UT Southwestern Medical Center was supported by a Shared Instrument Grant from NIH (1S10RR027972 to Q-X J). XZ is supported in part by grants from NIH (R01GM088197) and the Welch Foundation (I-1702). The crystallographic results shown in this report are derived from work performed at Argonne National Laboratory, Structural Biology Center at the Advanced Photon Source. Argonne is operated by U Chicago Argonne, LLC, for the US Department of Energy, Office of Biological and Environmental Research under contract DE-AC02-06CH11357. Part of this work was performed in laboratories constructed with support from NIH grant C06RR30414 (Dr Jerry Shay as the PI). ZJC is an investigator with the Howard Hughes Medical Institute.

## Additional information

### Competing interests

ZJC: Reviewing editor, *eLife*. The other authors declare that no competing interests exist.

### Funding

| Funder | Grant reference number | Author |
| --- | --- | --- |
| National Institutes of Health | R01 GM088745 | Hui Xu, Brian Borkowski, Qiu-Xing Jiang |
| National Institutes of Health | R01GM093271 | Hui Zheng, Brian Borkowski, Qiu-Xing Jiang |
| National Institutes of Health | R01GM088197 | Xiaojing He, Xuewu Zhang |
| National Institutes of Health | 1S10RR027972 | Qiu-Xing Jiang |
| American Heart Association | 12IRG9400019 | Qiu-Xing Jiang |
| Welch | I-1684, I-1389, and I-1702 | Hui Zheng, Qiu-Xing Jiang, Hui Xu, Zhijian J Chen, Xuewu Zhang |
| Cancer Prevention Research Institute of Texas | RP120474 | Hui Xu, Qiu-Xing Jiang |
| National Institutes of Health | R01 AI093967 | Zhijian J Chen |
| Howard Hughes Medical Institute | | Zhiheng Yu, Michael Jason de la Cruz, Zhijian J Chen |
| National Institutes of Health | C06RR30414 | Hui Xu, Hui Zheng, Lily J Huang, Brian Borkowski, Qiu-Xing Jiang |

The funders had no role in study design, data collection and interpretation, or the decision to submit the work for publication.

## Author contributions

HX, ZY, Acquisition of data, Analysis and interpretation of data, Drafting or revising the article; XH, HZ, LJH, Acquisition of data, Analysis and interpretation of data; FH, Approval of final published version, Acquisition of data, Contributed unpublished essential data or reagents; MJC, Acquisition of data, Analysis and interpretation of data, Contributed unpublished essential data or reagents; BB, Approval of the publication of this paper, Acquisition of data, Contributed unpublished essential data or reagents; XZ, ZJC, Conception and design, Analysis and interpretation of data, Drafting or revising the article; Q-XJ, Conception and design, Acquisition of data, Analysis and interpretation of data, Drafting or revising the article

## Additional files

### Supplementary files

• Supplementary file 1. Statistics for X-ray crystallographic data collection and model refinement.

### Major datasets

The following datasets were generated:

| Author(s) | Year | Dataset title | Dataset ID and/or URL | Database, license, and accessibility information |
|---|---|---|---|---|
| Zhang X, He X | 2014 | Crystal structure of horse MAVS card domain mutant E26R | 4O9L; http://www.rcsb.org/pdb/search/structidSearch.do?structureId=4o9l | Publicly available at RCSB Protein Data Bank (http://www.rcsb.org). |
| Zhang X, He X | 2014 | Crystal structure of horse MAVS card domain mutant R64C | 4O9F; http://www.rcsb.org/pdb/search/structidSearch.do?structureId=4o9f | Publicly available at RCSB Protein Data Bank (http://www.rcsb.org). |
| Jiang Q-X, Xu H | 2014 | cryoEM density map of MAVS CARD | EMD-5890; http://www.ebi.ac.uk/pdbe/entry/EMD-5890 | Publicly available at EMDB (http://www.ebi.ac.uk/pdbe/emdb/). |
| Jiang Q-X, Xu H | 2014 | cryoEM density map of MAVSdeltaProTM | EMD-5891; http://www.ebi.ac.uk/pdbe/entry/EMD-5891 | Publicly available at EMDB (http://www.ebi.ac.uk/pdbe/emdb/). |
| Jiang Q-X, Xu H | 2014 | Atomic model of MAVS CARD | 3J6C; http://www.rcsb.org/pdb/search/structidSearch.do?structureId=3j6c | Publicly available at RCSB Protein Data Bank (http://www.rcsb.org). |

The following previously published dataset was used:

| Author(s) | Year | Dataset title | Dataset ID and/or URL | Database, license, and accessibility information |
|---|---|---|---|---|
| Potter JA, Randall RE, Taylor GL | 2008 | Crystal structure of human IPS-1/MAVS/VISA/Cardif caspase activation recruitment domain | 2VGQ; http://www.rcsb.org/pdb/explore/explore.do?structureId=2vgq | Publicly available at RCSB Protein Data Bank (http://www.rcsb.org). |

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
