## [Decision Letter]

Thank you for sending your work entitled “Structural basis for the prion-like MAVS filaments in antiviral innate immunity” for consideration at *eLife*. Your article has been favorably evaluated by a Senior editor and 3 reviewers, one of whom, Wesley Sundquist, is a member of our Board of Reviewing Editors.

The Reviewing editor and the other reviewers discussed their comments before we reached this decision, and the Reviewing editor has assembled the following comments to help you prepare a revised submission. As you will see below, the editor and reviewers raise some questions as to the reliability of some of the key findings of the paper. As such, please respond to us by email in advance of a resubmission regarding how you propose to respond to these comments. This will allow the editors and reviewers to discuss your response, and provide you with feedback before you revise the manuscript.

The authors present a moderate-resolution cryo-EM reconstruction of the helical filaments formed by the CARD domain of MAVS and a lower-resolution reconstruction of filaments of the nearly intact cytoplasmic fragment that shows that the CARD domain polymer forms the core of the MAVS polymer. The reconstructions reveal that the CARD domain forms a left handed, three-stranded helix, and the CARD crystal structure has been docked into the 9.6 Å resolution map to produce a pseudoatomic model of the parallel triple helix and models for the different protein-protein interfaces that stabilize the structure. The model indicates that the helix is stabilized by two different types of interfaces, a hydrophilic intrastrand interface and a hydrophobic interstrand interface. A series of experiments support the accuracy and relevance of the structure, including extensive mutational analyses of the two major protein-protein interfaces and SR-SIM imaging of the tube-shaped MAVS polymers formed in cells.

The structure of the “activated” MAVS polymer is certainly of great general interest across multiple disciplines because this is the complex that responds to cytoplasmic antiviral sensors and initiates antiviral signaling cascades that lead to production of type I interferons and cytokines. The work is generally of high technical quality, and the extent of supporting experimentation is impressive. The work is therefore, in principle, appropriate for publication in *eLife*. Our two major concerns (described below) are: 1) it is difficult to be absolutely certain of the veracity of the reconstruction and pseudoatomic model, and 2) It is important to prove, unambiguously, that mitochondrial clustering and tube formation correlate with antiviral signaling and with the ability of MAVS to polymerize in vitro.

Critical points that must be addressed:

1) The CARD and MAVS(deltaTM) reconstructions appear to be carefully done and reasonable, but require additional validation. The correctness of the assigned symmetry and structure should be further demonstrated by providing supplemental figures that show: a) asymmetric maps with *no* symmetry imposed, and 2D class averages for the MAVS(deltaTM) analysis. b) Comparative residuals for the IHRSR experiments from different starting models (2-start, 3-start, and 4-start) and the power spectra for the translationally aligned boxes overlaid or displayed side-by-side with the power spectra calculated from projections of the final reconstructions. c) Experimental definition of the handedness of the polymers either by tilted images or rotary shadowing. d) A more extensive discussion of the resolution of the structure, e.g., does Figure 1—figure supplement 1 – the power spectrum of the final model – reveal layer lines beyond LL=9 at ∼1/17 Angstroms? Do lower contoured maps clearly show tubes of density for the different helices?

2) Figure 1 and Figure 1—figure supplement 1 do not adequately illustrate how well (or badly) the docked CARD domains fit into the map, and this is an important issue particularly because the CARD domains are globular, because the authors argue that the internal density requires a rearrangement of helices H2 and H5, and because there is unassigned density on the tube exterior. Ideally, this would be in the form of 3D movies that show the fit of the crystal structures into full and segmented density.

3) The SR-SIM experiments in Figure 5 lack a negative control, and this should be added (i.e., a reconstruction of a non-functional mutant such as the E26A mutation). Alternatively, the authors could test a more extensive set of mutants in the confocal experiments shown in Figure 5—figure supplement 1. The important issue is to prove, unambiguously, that mitochondrial clustering and tube formation correlate with antiviral signaling and with the ability of MAVS to polymerize in vitro.

---

## [Author Response]

*1) The CARD and MAVS(deltaTM) reconstructions appear to be carefully done and reasonable, but require additional validation. The correctness of the assigned symmetry and structure should be further demonstrated by providing supplemental figures that show: a) asymmetric maps with no **symmetry imposed, and 2D class averages for the MAVS(deltaTM) analysis. b) Comparative residuals for the IHRSR experiments from different starting models (2-start, 3-start, and 4-start) and the power spectra for the translationally aligned boxes overlaid or displayed side-by-side with the power spectra calculated from projections of the final reconstructions. c) Experimental definition of the handedness of the polymers either by tilted images or rotary shadowing. d) A more extensive discussion of the resolution of the structure, e.g., does*
Figure 1—figure supplement 1
*– the power spectrum of the final model – reveal layer lines beyond LL=9 at ∼1/17 Angstroms? Do lower contoured maps clearly show tubes of density for the different helices*?

These are excellent points.

a) It was our omission that in the original submission we did not present more meta-data for the analysis of the MAVS(delta)TM (now called MAVS delta ProTM) filaments. The revised version has the 2D class averages and two views of the 3D map before symmetry imposition in Figure 4—figure supplement 1 respectively. We also calculated the power spectrum from the 360 projections of the final 3D reconstruction and compared it with that of the dataset in Figure 4—figure supplement 1.

b) In the raw dataset the meridional line (layer line 9) was visible, which provided a reliable estimate of the axial rise, and a decoupling of the two helical parameters in our search for the right ones. We tested 4-start helical symmetry previously and were not able to converge to the right axial rise. We started with different rotational angles from 40 to 51 degrees, and did not find one that works for n=4. One example is now shown in Figure 1—figure supplement 1. We therefore ruled out the possibility of n=4. Based on the indexing of the layer lines and peaks, we did not find that a 2-start model was a possibility. As the reviewers requested, we tested the 2-start helical symmetry, but the refinement did not converge and the axial rise went continuously from the correct 16.7 Å to the incorrect 19.5 Å after 50 runs. The reviewers asked about the calculations of residuals with “xhelicals”. We looked into it. In the new distribution of the IHRSR, this program was neither recompiled nor tested. When we used it to search for symmetry, it did not produce the right results as the well-tested “hsearch_lorentz”. We therefore did not use it in our analysis. But this does not change our conclusion that n=3 is the right rotational symmetry.

In the revision, we have compared the power spectra of the raw dataset and the averaged power of the 360 projections from the final 3D map. These are now in Figure 1—figure supplement 1 and Figure 4—figure supplement 1.

c) The reviewers asked for independent verification of the handedness of the filament. We conducted electron cryo-tomography (cryo-ET) and freeze-etch experiments of the frozen filaments. The data by freeze-etch EM are not available due to the delayed scheduling of the experiments to be done out of our campus. The cryo-ET data are presented in Figure 1—figure supplement 2. It showed fairly conclusively that the filaments are indeed left-handed.

d) A separate evaluation of the resolution is an excellent idea. As suggested, we now displayed the averaged power spectrum from 360 projections calculated from the 3D cryoEM map rotated in 1-degree step along the helical axis. Our rationale for this test is based on the fact that the helical filament is a 1D crystal, and the layer lines should extend horizontally to the resolution range we estimated from FSC. In Figure 1—figure supplement 1, the right panel showed the extension of the layer lines horizontally up to ∼8.2 angstroms (red circle). We used the SQUARE root function (modulus of the structural factors) to enhance the contrast of the weak lines here.

In ‘Video 2’, we made a separate presentation at a higher contour level to show that some rod-shaped densities overlap with specific alpha-helices that were used to guide our structural docking in Figure 1 and Figure 1—figure supplement 2.

*2)*
Figure 1
*and*
Figure 1—figure supplement 1
*do not adequately illustrate how well (or badly) the docked CARD domains fit into the map, and this is an important issue particularly because the CARD domains are globular, because the authors argue that the internal density requires a rearrangement of helices H2 and H5, and because there is unassigned density on the tube exterior. Ideally, this would be in the form of 3D movies that show the fit of the crystal structures into full and segmented density*.

Thanks for this excellent suggestion. We generated two 3D movies. The first one shows the cryoEM map at different orientations and the segmentation of one unit. The second illustrates the docking of the CARD X-ray structural model into the density map. It also shows the positioning of specific alpha-helices (H1, H4 and H3 as well as H6) into the density of one unit, the small discrepancies for H2 and H5 at the side facing the inner pore of the MAVS CARD filament, and the rod-shaped densities at slightly higher threshold, which overlap with H1, H4 and H6. The last part of Video 2 also presented the assembly of individual CARD domains into a helical filament.

*3) The SR-SIM experiments in*
Figure 5
*lack a negative control, and this should be added (i.e., a reconstruction of a non-functional mutant such as the E26A mutation). Alternatively, the authors could test a more extensive set of mutants in the confocal experiments shown in*
Figure 5—figure supplement 1*. The important issue is to prove, unambiguously, that mitochondrial clustering and tube formation correlate with antiviral signaling and with the ability of MAVS to polymerize in vitro*.

This is a very important point. We have addressed it from five different aspects.

First, we prepared *Mavs-null* MEFs stably expressing an extensive set of MAVS mutants. The confocal images of more mutants (E26A, E80A, F16A, F16H and Y30A) are now in Figure 5—figure supplement 1 and Figure 5—figure supplement 2. E80A serves as a control for the electrostatic interaction and E26A is a loss-of-function mutation at the inter-strand electrostatic interface. The formation of bright puncta of MAVS correlated well with the interferon production assay in Figure 2. For the intra-strand interface, F16A and Y30A are two loss of function mutants and F16H is functional as we found in Figure 3. The 4-color confocal images also allowed us to evaluate the correlation between the formation of bright MAVS puncta and the nuclear translocation of p65 (the signaling of MAVS). These data support strong correlations among the filament formation through the two identified interfaces, the MAVS puncta formation in cells and the MAVS signaling (through the activation of NF-κB).

Second, as the reviewers suggested, we statistically analyzed the percentage of cells with p65 nuclear translocation (positive MAVS signaling) that displayed the formation of large bright MAVS puncta using *Mavs-null* MEFs reconstituted with wild-type MAVS. Among cells that were positive in p65 nuclear translocation, more than 70% of them had strong MAVS puncta formation in confocal imaging. The new data are presented in Figure 5—figure supplement 3. They strike a statistically significant and strong correlation between puncta formation and MAVS signaling.

Third, the mutants used for the confocal experiments were tested for their capability of rescuing MAVS signaling in the *Mavs-null* MEFs. q-RT-PCR experiments showed that the viral infection induced IFN-beta production in those cells correlated well with their ability (or inability) to form bright puncta in the confocal imaging. These data are in Figure 5—figure supplement 3.

Fourth, based on the results in Figure 2 and Figure 3 in the previous version, we have altogether established *Mavs-null* MEFs stably expressing E26A, R64A, R65A, F16A, Y30A as the loss-of-function mutations, and E80A, L48A, F16H, W56Y, Y30F and R77A as negative controls. We utilized SDD-AGE to analyze the formation of MAVS aggregates from mitochondrial extracts in those cells after viral infection. The results agree well with the other assays, and are presented in Figure 5—figure supplement 3. They argue strongly the tight coupling between the CARD filament formation and the MAVS aggregate formation in virus-infected cells.

Finally, as the reviewers suggested, in addition to the mutants that we have tested in Figure 2 and Figure 3 in the previous version, we have tested almost all conserved charged residues for their effects on virus-induced IFN-beta production in cells. Mutations of the surface-exposed residues (R52A, E70A, R77A, E80A, D86A and E87A) that are not involved in the inter-strand interaction did not impair MAVS signaling. On the other hand, R41, R43 and D40, which are close to the positively charged patch of R64 and R65 at the electrostatic interface (inter-strand), inhibited MAVS signaling. Their effects and their positions in the helical filament are added to Figure 2 and Figure 2. These data are now rather complete and suggest that the electrostatic interaction between strands have relatively high specificity. It argues strongly that the filament formation through the electrostatic interface is critically correlated with the MAVS signaling (interferon production).

We believe that these new data together with those in the first submission lever compelling evidence that in virus-infected cells the MAVS puncta formation, MAVS antiviral signaling and MAVS polymerization are tightly coupled.